# Chance-constrained Flow Matching for High-Fidelity Constraint-aware Generation

## Abstract

Generative models excel at synthesizing high-fidelity samples from complex data distributions, but they often violate hard constraints arising from physical laws or task specifications. A common remedy is to project intermediate samples onto the feasible set; however, repeated projection can distort the learned distribution and induce a mismatch with the data manifold. Thus, recent multi-stage procedures attempt to defer projection to "clean" samples during sampling, but they increase algorithmic complexity and accumulate errors across steps. This paper addresses these challenges by proposing a novel *training-free method*, *Chance-constrained Flow Matching* (CCFM), that integrates stochastic optimization into the sampling process, enabling effective enforcement of hard constraints while maintaining high-fidelity sample generation. Importantly, CCFM guarantees feasibility in the same manner as conventional repeated projection, yet, despite operating directly on noisy intermediate samples, it is theoretically equivalent to projecting onto the feasible set defined by clean samples. This yields a sampler that mitigates distributional distortion. Empirical experiments show that CCFM outperforms current state-of-the-art constrained generative models in modeling complex physical systems governed by partial differential equations and molecular docking problems, delivering higher feasibility and fidelity.

## 1 Introduction

Diffusion and flow-matching generative models have achieved remarkable performance across domains, including image generation (Ho et al., 2020; Lipman et al., 2022), robotic motion planning (Zhang & Gienger, 2024; Liang et al., 2025), physical system modeling (Cheng et al., 2024; Utkarsh et al., 2025), and other scientific applications (Yim et al., 2024; Miller et al., 2024; Christopher et al., 2025). Although these models are highly effective at producing content that follows complex distributions, a key challenge is ensuring that generated samples strictly satisfy specific constraints or physical laws. For example, physical systems modeling should adhere to partial differential equations, or designing protein-ligand docking must satisfy steric and chemical constraints.

To address these challenges, a growing literature has started exploring constrained handling for generative models. However, these approaches present several notable limitations: *(i)* First, gradient-based guidance methods are dominant in many scientific applications, but, while elegant, such methods cannot guarantee the feasibility of the imposed constraints (Huang et al., 2024); *(ii)* Second, repeated projection methods, on the other hand, can enforce strict feasibility, but often at the expense of sample quality (Christopher et al., 2024; Santos et al., 2025); *(iii)* Finally, a more recent line of work attempts to mitigate these issues via an *extrapolation–correction–interpolation* (ECI) procedure (Cheng et al., 2024; Utkarsh et al., 2025), which reduces sample distortion. However, the performance of these methods strongly depends on the accuracy of the one-step extrapolation step and constraint formulation, leaving the integration of hard constraints during sampling largely ad hoc and often accompanied by increased overhead, without a unifying theoretical foundation.

To address these challenges, this paper introduces *Chance-constrained Flow Matching* (CCFM), a training-free framework for constrained generation that guarantees feasibility while maintaining high-fidelity sample generation. Fidelity here goes beyond feasibility, capturing closeness to the target distribution, such as realistic bound structures in molecular docking or low mean squared error against ground-truth solutions in PDEs. The proposed ideas rely on a key observation: under

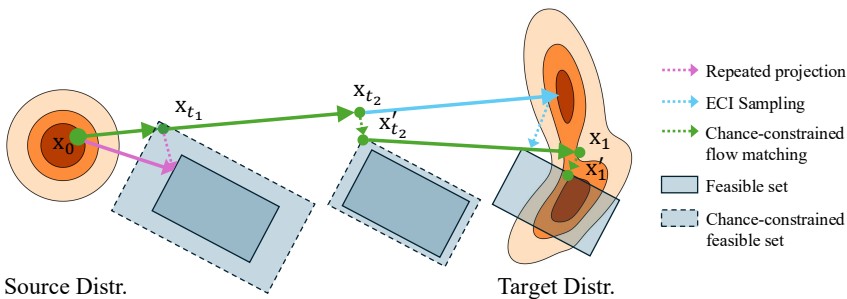

Figure 1: Overview of chance-constrained flow matching (CCFM). Solid arrows follow the sampling path; dashed arrows indicate projection steps. In CCFM, early states remain unchanged ($x_{t_1}$), intermediate states are minimally corrected ($x_{t_2} \to x'_{t_2}$), and the final state is guaranteed feasible ($x'_1$) in a high-probability region. Repeated projection distorts the sampling path. ECI sampling relies on the quality of one-step prediction and may project outside high-probability regions.

the standard flow matching parametrization using the optimal transport (OT) path, each intermediate (noisy) state can be expressed in an affine relation to the initial state (noise) and the clean terminal sample (Liu et al., 2022; Lipman et al., 2024). This allows us to make a novel connection between flow matching and chance-constrained programming (CCP) Kall et al. (1994), an optimization framework widely used in operations research to enforce constraints with probabilistic guarantees. Modeling the initial state as a random variable and posing the constraint enforcement as a chance-constrained program allows us to derive a feasible set on noisy samples that is theoretically equivalent to the feasible set defined on clean samples (for a chosen risk level $\alpha$). This enables direct projection of intermediate states, avoiding extrapolation used in ECI sampling, while remaining theoretically equivalent to projection on the clean sample (with probability $1 - \alpha$) and thus preserving feasibility. A schematic illustration of the idea is provided in Figure 1.

**Contribution.** This work makes the following key contributions: **(1)** It introduces *Chance-constrained Flow Matching* (CCFM), a novel framework that casts the projection step from a stochastic optimization perspective, enabling concise and effective enforcement of hard constraints while preserving the high-fidelity sample generation. **(2)** It establishes a theoretical framework for enforcing hard constraints in the sampling process of generative models through the perspective of chance-constrained programming. The analysis demonstrates that CCFM guarantees feasibility while preserving sample quality. **(3)** It demonstrates the effectiveness of CCFM across two scientific domains: molecular docking, specifically modeling the structural transformation between unbound and bound protein conformations, and modeling physical systems governed by PDEs, including Reaction–Diffusion and Navier–Stokes systems. In both domains, CCFM produces samples that satisfy complex constraints and physical principles, achieving new state-of-the-art performance.

## 2 RELATED WORK

Recent progress on constrained generative modeling seeks to modify the training or inference stages of these models to account for constraints underlying the data-generating processes. Three predominant strategies can be identified: **(1)** *Penalty Training* introduces constraints violation penalties into the training objectives (Naderiparizi et al., 2025; Li et al., 2025; Baldan et al., 2025). While simple, such penalties rarely deliver hard guarantees and can be brittle in out-of-distribution settings. **(2)** *Classifier guidance* incorporates an additional penalty term during sampling (Huang et al., 2024; Jacobsen et al., 2024). These methods offer much flexibility (as surrogate models can be used in place of constraints), but provide no feasibility guarantees and, additionally, degrade under complex constraint structures (e.g., nonlinear or nonconvex constraints). **(3)** Finally, *Projection-based methods* enforce feasibility by projecting samples onto the feasible mainfold (Christopher et al., 2024; Liang et al., 2025). However, these projections are applied at every step of the generative process, and when projections are performed in the early, noisy, states, the process has been shown to substantially distort the training data manifold (Chung et al., 2022; Santos et al., 2025), as the projection disrupts the exactness of sampling dynamics. To mitigate this distortion issue, Cheng et al. (2024) and Utkarsh et al. (2025) develop ECI-style sampling methods that apply projection only at the final

clean sample through a one-step extrapolation step. This, however, comes with an increased sampling complexity cost and, as we will show in our experiments, introduces additional errors from final-state prediction and interpolation.

In contrast, the proposed *CCFM* exploits the optimal transport path in flow matching to enable direct projection at intermediate states, mitigating degradation from projection methods and the complexity of ECI sampling, while ensuring feasibility and high-fidelity generation.

## 3 PRELIMINARIES

Prior to introducing the proposed CCFM approach, the paper starts by reviewing the concepts of generative flow matching and chance-constrained programming.

**Flow matching.** Flow matching (FM) is a generative framework that learns a time-dependent velocity field $u_t : [0, 1] \times \mathbb{R}^d \to \mathbb{R}^d$, which defines a mapping (*flow*) $\psi_t : [0, 1] \times \mathbb{R}^d \to \mathbb{R}^d$ that transforms samples $x_0$ from a source distribution $p$ into samples $x_1 = \psi_1(x_0)$ on a target distribution $q$ via solving the ordinary differential equation (ODE) (Lipman et al., 2024):

$$\frac{d}{dt}\psi_t(x) = u_t(\psi_t(x)), \quad \text{with initial condition} \quad \psi_0(x) = x. \tag{1}$$

Here, the variable $t \in [0, 1]$ represents the time in FM, with $t = 0$ corresponding to the initial state (noise) and $t = 1$ to the final state (clean sample). To learn the velocity field, a probability path $\{p_t\}_{t \in [0,1]}$ is defined to connect $p_0 = p$ and $p_1 = q$. This reduces distribution matching to a supervised learning problem, as the path provides intermediate states corresponding to target velocities for training $u_t$. The optimal transport (OT) displacement path is widely adopted as it ensures smooth and cost-minimizing interpolations between distributions that yield stable intermediate states for training (Liu et al., 2022; Kerrigan et al., 2023; Kornilov et al., 2024). The quadratic-cost OT displacement path is implemented conditionally on data $X_1 \sim q$ by mixing with a simple source $X_0 \sim p$ (e.g., $p = \mathcal{N}(0, I)$) (McCann, 1997; Villani et al., 2008; Villani, 2021):

$$X_t = \psi_t(X_0 | X_1) = (1 - t) X_0 + t X_1, \qquad t \in [0, 1]. \tag{2}$$

The training loss is given by: $\mathcal{L}_{\text{CFM}}(\theta) = \mathbb{E}_{X_0 \sim p, X_1 \sim q, t \sim \text{Unif}[0,1]} \|u_\theta(X_t, t) - (X_1 - X_0)\|^2$, where $u_\theta$ is the learnable velocity field parameterized by $\theta$, and $(X_1 - X_0)$ denotes the target velocity. After training, the initial state is drawn by $x_0 \sim p$ and propagated through the flow ODE $\frac{d}{dt}x_t = u_\theta(x_t, t)$, which can be solved with the Euler method (Iserles, 2009):

$$x_{t+\Delta t} = x_t + \Delta t \, u_\theta(x_t, t). \tag{3}$$

If $u_\theta$ matches the ground-truth marginal velocity, the induced flow transports $p$ to $q$. In deterministic settings, where $X_1$ is determined by $X_0$, the trajectories are linear (Lipman et al., 2022; 2024):

$$x_t = (1 - t) x_0 + t x_1. \tag{4}$$

**Chance-constrained Programming.** In operations research and stochastic optimization, *chance-constrained programming* (CCP) (Kall et al., 1994) formalizes decision-making under uncertainty by requiring constraints to hold with a specific probability, rather than for every realization of the uncertainty. Consider a general optimization problem with objective function $f(\cdot)$, constraints $g(\cdot)$, and decision variable $x$:

$$\min_y f(x) \quad \text{s.t.} \quad g(x) \leq 0. \tag{5}$$

When the constraints depend on a random input $\xi$, i.e., $g(x, \xi) \leq 0$, it is typically intractable to enforce feasibility for all realizations of $\xi$. To address this, CCP imposes a probabilistic feasibility requirement (Nemirovski & Shapiro, 2007; Liang et al., 2024):

$$\min_y f(x) \quad \text{s.t.} \quad \mathbb{P}\big(g(x, \xi) \leq 0\big) \geq 1 - \alpha, \tag{6}$$

where $\alpha \in (0, 1)$ is a user-specified risk level controlling the allowable constraint violation probability. In this way, the solution to (Problem 6) ensures that the constraints are satisfied with probability at least $1 - \alpha$ under the uncertainty in $\xi$. As $\alpha \to 0$, CCP approaches exact feasibility. Variants and relaxations (e.g., convex approximations or scenario-based surrogates) are standard in Operations Research practice when Problem 6 is nonconvex or intractable, but the core modeling principle remains the probabilistic guarantee.

## 4 METHODOLOGY

This section proposes CCFM, a novel integration of flow matching with chance-constrained programming to generate high-fidelity samples with feasibility guarantees. We begin by revisiting the classical repeated projection method. Then, we investigate how to incorporate chance constraints into flow matching, followed by an equivalent tractable reformulation that enables efficient implementation. Finally, we provide a theoretical analysis that examines feasibility guarantees and shows equivalence to repeated projection and ECI sampling.

**Repeated Projection.** Flow matching evolves an initial noise sample toward a target sample via the learned vector field using explicit Euler update (see Equation 3). To enforce feasibility of the generated samples, several works apply *repeated projections* (Christopher et al., 2024; Liang et al., 2025). Let $\mathcal{C} = \{x : g(x) \leq 0\}$ denote thes feasible region. At each step, a projection operator is introduced to map intermediate samples onto $\mathcal{C}$:

$$\mathcal{P}(x_t) = \arg\min_{x'_t \in \mathcal{C}} \|x'_t - x_t\|_2^2, \tag{7}$$

where $x'_t$ is the nearest feasible point satisfying the constraints. With this operator, the (projected) sampling rule Equation 3 becomes

$$x_{t+\Delta t} = \mathcal{P}(x_t + \Delta t \, u_\theta(x_t, t)), \tag{8}$$

that is, each tentative Euler step is followed by a projection onto $\mathcal{C}$. This projected-Euler scheme preserves per-step feasibility (when $\mathcal{C}$ is convex).

### 4.1 CHANCE-CONSTRAINED FLOW MATCHING

Observe that the projection operator $\mathcal{P}(\cdot)$ is applied to intermediate noisy samples, while the constraint set is defined with respect to the clean sample. As discussed earlier, this mismatch has been shown to alter the sampling trajectory relative to the learned flow, potentially distorting the induced distribution (Kynkäänniemi et al., 2024; Rojas et al., 2025). To address this challenge, *Chance-constrained Flow Matching* reformulates Problem 8 via a stochastic optimization lens which allows us to adaptively adjust the constraints tightness over the sampling time $t$. Although we formulate the constraints at each sampling time $t$ in a probabilistic form, the constraints we want to enforce remain the original deterministic data-level constraints.

To do so, we exploit a key property of FM: the linearity of the flow under the OT path, as shown in Equation 4. This linear structure allows us to characterize the dependency between the clean final state $x_1$, any intermediate state $x_t$, and the initial noise $x_0$. In particular, the final state $x_1$ can be expressed in closed form as a function of $x_t$ and $x_0$ by rearranging Equation 4 as

$$x_1 = t^{-1}x_t - (1-t)t^{-1}x_0. \tag{9}$$

Since the constraints are defined on the clean sample $x_1$, we substitute Equation 9 into constraints $g(x_1) \leq 0$, yielding

$$g(t^{-1}x_t - (1-t)t^{-1}x_0) \leq 0.$$

Note that, because $x_0$ is sampled from the noise distribution, the term $(1-t)t^{-1}x_0$ is random. This allow us to introduce a random variable $\xi = (1-t)t^{-1}x_0$ which isolates the stochastic component, and rewrite the constraint compactly as

$$g(t^{-1}x_t - \xi) \leq 0.$$

This reformulation isolates the source of uncertainty explicitly and naturally motivates a chance-constrained treatment at each time step $t$:

$$\mathbb{P}_\xi(g(t^{-1}x_t - \xi) \leq 0) \geq 1 - \alpha, \tag{10}$$

where $\xi := (1-t)t^{-1}x_0$ and $\mathbb{P}_\xi$ denotes probability with respect to the distribution of $\xi$. The corresponding chance-constrained projection operator is then defined as the closest feasible adjustment of $x_t$ that satisfies the probabilistic constraints:

$$\mathcal{P}_{cc}(x_t, 1-\alpha) = \arg\min_{x'_t} \|x'_t - x_t\|_2^2 \quad \text{s.t.} \quad \mathbb{P}_\xi(g(t^{-1}x_t - \xi) \leq 0) \geq 1 - \alpha. \tag{11}$$

Then, after embedding the projection operator $\mathcal{P}_{cc}$ after each sampling step, the update rule for the CCFM sampling process is given by

$$x_{t+\Delta t} = \mathcal{P}_{cc}\left(x_t + \Delta t\, u_\theta(x_t, t), \phi(t)\right). \tag{12}$$

where $\phi(t) = (t/2)^n$ serves as a probabilistic scheduler that adaptively modulates the satisfaction, where $n$ is a user-specified value. As the sampling time $t$ increases, the contribution of $x_0$ to $x_t$ decreases, while the probability of satisfaction of the constraints increases simultaneously.

These effects yield an adaptive tightening of the constraints, and as $t \to 1$, the random variable $\xi$ vanishes, leading the chance-constrained projection operator to degenerate into its deterministic counterpart. The detailed procedure of CCFM is summarized in Algorithm 1.

---
**Algorithm 1:** Chance-constrained Flow Matching

**Input:** Learned vector field $u_\theta$, Euler steps $N$.

Sample noise function $x_0 \sim p.$;

**for** $t \leftarrow 0, 1/N, 2/N, \ldots, (N-1)/N$ **do**

    $x_{t+1/N} \leftarrow x_t + u_\theta(x_t, t)/N$;

    $x_{t+1/N} \leftarrow \mathcal{P}_{cc}(x_{t+1/N}, \phi(t))$;

**return** $x_1$;

---

**Remark 1.** CCFM can be contrasted with existing projection strategies. *Repeated projection* enforces the same constraints at every step. When the constraints are too tight, especially early in sampling, the projection may significantly distort the learned velocity. This deviation from the learned distribution reduces generation quality. On the other hand, at each step $t$, *ECI* predicts the final state in one step, projects it, and interpolates back to the intermediate state. Although it avoids perturbing noisy samples, the velocity depends on a projected one-step prediction rather than the learned distribution, making performance sensitive to prediction quality. In contrast, *CCFM* adaptively adjusts constraint tightness over time $t$. This yields minimal modifications of the learned velocity toward feasibility and enables high-fidelity generation without reliance on *one-step predictions*.

## 4.2 TRACTABLE REFORMULATION FOR CCFM

While the chance-constrained projection operator is conceptually appealing, directly optimizing over a probabilistic constraint is generally intractable. Fortunately, the source distribution used in FM is typically a simple distribution (e.g., Gaussian); this structure enables a tractable reformulation of the chance-constrained projection operator into an equivalent deterministic problem. Such reformulations are well studied in the optimization communities Nemirovski & Shapiro (2007). The following proposition establishes this tractable reformulation:

**Proposition 1** (Tractable reformulations of chance-constrained projection operator)**.** *Let $t \in (0, 1]$ and fix a risk level $\alpha \in (0, 1)$. Consider $a, x_t \in \mathbb{R}^d$, $b \in \mathbb{R}$, and $\xi \sim \mathcal{N}(0, \sigma^2 I_d)$. In particular, when $\xi = (1-t)t^{-1}x_0$ with $x_0 \sim \mathcal{N}(0, I_d)$ (the standard linear interpolation path), one has $\sigma = \frac{1-t}{t}$. Denote the Euclidean norm by $\|\cdot\|_2$, and let $z_{1-\alpha} := \Phi^{-1}(1-\alpha)$ and $z_{1-\alpha/2} := \Phi^{-1}(1-\alpha/2)$, where $\Phi$ is the standard normal CDF. Then:*

   i. **Linear constraints**: *For $g(x) := a^\top x - b$, the chance constraint $\mathbb{P}_\xi\big(g(t^{-1}x_t - \xi) \le 0\big) \ge 1 - \alpha$ is equivalent to the deterministic constraint: $a^\top x_t \le t\, b - t\, \sigma\, \|a\|_2\, z_{1-\alpha}$.*

   ii. **Quadratic constraints**: *For $g(x) := (a^\top x)^2 - b$ with $b > 0$, the chance constraint $\mathbb{P}_\xi\big((a^\top(t^{-1}x_t - \xi))^2 \le b\big) \ge 1 - \alpha$ is enforced by $|a^\top x_t| \le t\left(\sqrt{b} - \sigma\, \|a\|_2\, z_{1-\alpha/2}\right)$, if $\sqrt{b} \ge \sigma\, \|a\|_2\, z_{1-\alpha/2}$.*

Since $z_{1-\alpha} = \Phi^{-1}(1-\alpha)$ is monotone in $\alpha$ and the variance $\sigma = \frac{1-t}{t}$ decreases as $t \to 1$, the effective constraint bound tightens progressively with the sampling time $t$. The paper reports proofs for all theoretical results in Appendix A.

## 4.3 THEORETICAL ANALYSIS FOR CCFM

Having established a tractable chance-constrained projection for any time $t \in (0, 1]$, the next result shows that this procedure guarantees the feasibility of generated samples.

**Corollary 2** (Feasibility Guarantee)**.** *At the sampling time $t = 1$, the chance-constrained projection operator $\mathcal{P}_{cc}$ from Proposition 1 reduces to the standard Euclidean projection onto the deterministic*

*feasible set* $\mathcal{F}_{\text{det}} := \{x' \in \mathbb{R}^d \mid g(x') <= 0\}$. *That is,*

$$\mathcal{P}_{\text{cc}}(x_1, 1 - \alpha) = \operatorname*{arg\,min}_{x_1' \in \mathcal{F}_{\text{det}}} \|x_1' - x_1\|_2^2.$$

Proposition 1 and Corollary 2 indicate that for $t < 1$ the operator provides a probabilistic guarantee, while at $t = 1$ it degenerates to the deterministic projection operator, thereby ensuring feasibility. Next, we demonstrate that feasibility at the final state propagates backward along the linear OT path, and show that the projection operators at intermediate times admit an explicit geometric relation to the projection on final clean samples. All the results below assume $g : \mathbb{R}^d \to \mathbb{R}$ proper, lower semicontinuous, and convex, as recurrent in optimization theory. We define the clean feasible set as $\mathcal{C}_1 := \{x_1 \in \mathbb{R}^d : g(x_1) \leq 0\}$, which is assumed to be nonempty. Finally, for $t \in (0, 1]$, we assume that the sampling path is $x_t = (1 - t)x_0 + tx_1$, and we use the affine map $M_t(x) := \frac{\left(x - (1-t)x_0\right)}{t}$.

**Theorem 3** (Pathwise feasibility propagation)**.** *Under the asssumptions above, the probability* $\mathbb{P}_\xi$ *w.r.t. the degenerate law concentrated at $\xi$ of satisfying the imposed constraints at time $t \in (0, 1]$ is:*

$$\mathbb{P}_\xi\big(g\big(t^{-1}x_t - \xi\big) \leq 0\big) = \begin{cases} 1, & \text{if } g(x_1) \leq 0, \\ 0, & \text{otherwise,} \end{cases} \tag{13}$$

*where* $\xi := \frac{1-t}{t}x_0$. *In particular, the chance constraint* 10 *holds if and only if $x_1 \in \mathcal{C}_1$.*

**Theorem 4** (Geometric structure of the samplewise feasible sets)**.** *For every $t \in (0, 1]$,*

$$\mathcal{C}_t(x_0) = (1 - t)x_0 + t\,\mathcal{C}_1,$$

*where* $\mathcal{C}_t(x_0) := \{x_t : g(M_t(x_t)) \leq 0\}$ *and* $\mathcal{C}_1 := \{x_1 : g(x_1) \leq 0\}$. *Then, let $\mathbb{P}_t$ and $\mathbb{P}_1$ denote Euclidean projections onto $\mathcal{C}_t(x_0)$ and $\mathcal{C}_1$, respectively, for any $x \in \mathbb{R}^d$,*

$$\boxed{\mathbb{P}_t(x) = (1 - t)x_0 + t\,\mathbb{P}_1\big(M_t(x)\big),} \tag{14}$$

*and the minimizer is unique.*

The above is a key result for this work, as it offers a key geometric insight. It implies that projecting onto the time-varying feasible set $\mathcal{C}_t(x_0)$, for any initial state $x_0$, can be geometrically decomposed: it is equivalent to first mapping the point to the final state via $M_t$, then performing a single projection onto $\mathcal{C}_1$, and finally mapping the result forward. *Crucially, this shows that the geometry of the feasible space is preserved throughout the generation process.*

**Remark 2.** Notice that, while it is customary in optimization to assume convexity in the constraint spaces to prove theoretical results, empirically, our results will show that these guarantees hold up in highly non-convex cases. Appendix B details the chance-constraint implementation for both convex and non-convex constraints adopted in our experiments.

**Remark 3.** In practice, numerical discretization may introduce slight curvature around the ideal straight-line characteristics in Equation 4. Nevertheless, recent theoretical advances show that the ground-truth velocity field can be approximated using Lipschitz neural networks, with approximation error bounded by $(\sqrt{d} + 1)\epsilon$ for arbitrary $\epsilon > 0$, where $d$ is the dimension of data (Zhou & Liu, 2025). This guarantee ensures that the discrepancy between the learned and ground-truth velocity fields remains small. Moreover, although an approximation error exists, Gao et al. (2024) show that both estimation and discretization errors are controllable: the velocity estimation error decays at a rate $\tilde{O}((nt^2)^{-1/(d+3)})$, where $n$ is the number of training samples and $\tilde{O}$ hides polylogarithmic factors. The discretization error scales as $O(\sqrt{\Upsilon})$, thus smaller step sizes $\Upsilon$ directly reduce error at a square-root rate. The use of straight-line characteristics has also been adopted in prior works with strong empirical performance (Liu et al., 2022; 2023), and our experiments further corroborate their effectiveness.

## 5 EXPERIMENTS

This section evaluates CCFM against state-of-the-art constrained generative models and shows its generality on two tasks in diverse scientific domains: **Molecular Docking** and **PDE Solution Generation**. We focus on two key aspects of constrained generation: (i) satisfaction of hard constraints

and (ii) sample fidelity. Further details on the problem formulation, constraint enforcement methods, dataset, and model specifications are provided in Appendix B. Sensitivity analysis and additional experimental results are presented in Appendix C.

## 5.1 EXPERIMENTAL SETTING

**Molecular Docking.** The task is to predict ligand poses within protein pockets while respecting geometric and physical constraints. Experiments are performed on the PDBBind docking benchmark (Liu et al., 2017), which provides a standard testbed for evaluating protein–ligand docking methods. Comparisons are made against a *repeated projection (RP)* baseline and *FlexDock* (Corso et al., 2025), the current state-of-the-art docking framework, which enforces structural constraints by embedding penalty terms during training and further ensuring validity at inference through rejection sampling.

The evaluation also focuses on: **(1)** *Fidelity*, which is quantified by ligand RMSD (L-RMSD) (fractions below 1Åand 2Å) and all-atom RMSD (A-RMSD) (fractions below 2Å). **(2)** *Feasibility*, which is assessed by PoseBusters validity (PB Valid) (Buttenschoen et al., 2024) and its conjunction with RMSD thresholds, measuring the physical plausibility of predicted complexes.

**PDE Solution Generation.** Next, following (Utkarsh et al., 2025), we consider a generative task on two representative PDE systems: the 1-D *reaction–diffusion equation*, subject to linear initial conditions (IC) and nonlinear global mass conservation laws (CL), and the 2-D *Navier–Stokes equation*, subject to linear initial conditions (IC) and linear conservation laws (CL). The reaction–diffusion problem is discretized on a $(128 \times 100)$ space–time grid, while the Navier–Stokes problem is posed on a $64 \times 64 \times 50$ grid. Unlike (Cheng et al., 2024; Huang et al., 2024), which evaluate on only 10 test initial conditions, we adopt 90 unseen (diverse) conditions to yield a challenging and discriminative benchmark.

Comparisons are conducted against several representative baselines: (a) *Physics-Constrained Flow Matching (PCFM)*(Utkarsh et al., 2025), which is the current state of the art; it extends ECI sampling (Cheng et al., 2024) to nonlinear constraints and reduces to ECI in the linear setting; (b) *DiffusionPDE* (Huang et al., 2024), which enforces physical priors through penalty terms during iterative sampling; (c) *Functional Flow Matching (FFM)* (Kerrigan et al., 2023), a pretrained unconstrained FM. All methods employ the same pretrained FFM backbone and are evaluated with identical sampling steps, differing only in how constraints are embedded into the sampling process; and (d) a *Repeated Projection (RP)* approach, which naively projects the state onto the feasible set at each sampling step.

Performance is evaluated along two complementary dimensions: **(1)** *Feasibility* is quantified by the constraint violation (CV), which measures deviations from the imposed boundary, initial, and conservation constraints. **(2)** *Fidelity* is evaluated using the pointwise mean squared error of the mean (MMSE) and the pointwise mean squared error of the standard deviation (SMSE), capturing distributional alignment in terms of first- and second-order statistics.

**Physics-informed motion.** Following the setting introduced in (Christopher et al., 2024), this task examines whether a generative model can synthesize video sequences that respect physical rules. The objective is to generate multi-frame trajectories of a moving object whose positions must satisfy discrete ordinary differential equation (ODE) updates coupling the position and velocity across time steps. Training data include only Earth-gravity motion, while test cases with Moon gravity provide an out-of-distribution evaluation. Comparisons are conducted against: (a) an *unconstrained* flow model, which follows the standard sampling process without any constraint enforcement; (b) a *repeated projection (RP)* baseline, which projects each intermediate sample onto the constraint set.

Evaluation focuses on two aspects: (1) Feasibility, measured by the violation of the ODE constraints across frames; (2) Fidelity, assessed by the Fréchet Inception Distance (FID), reflects the visual quality of generated sequences under both Earth and Moon gravity.

## 5.2 PERFORMANCE EVALUATION FOR MOLECULAR DOCKING

Table 1 reports docking results on the PDBBind docking benchmark under varying numbers of generated samples and inference steps. Notice how, across all configurations, *CCFM* consistently

Table 1: PDBBind docking results. #S denotes the number of samples generated per complex, with the best pose selected for evaluation; #Stp is the number of inference steps. L-RMSD/A-RMSD report fractions of complexes below the given thresholds, PB Valid is the proportion passing Pose-Busters checks, with +L and +A further requiring L-RMSD< 2Å or A-RMSD< 2Å, respectively.

| #S | #Stp | Methods | L-RMSD (%) | | A-RMSD (%) | PB Valid (%) | | | Running |
|---|---|---|---|---|---|---|---|---|---|
| | | | < 1Å | < 2Å | < 2Å | All | +L < 2Å | +A < 2Å | Time (s) |
| 1 | 2 | CCFM | **8.19** | **32.20** | **74.01** | **47.74** | **22.03** | **35.88** | 0.64 |
| | | FlexDock | 7.06 | 31.36 | 73.73 | 20.34 | 9.32 | 14.97 | 0.52 |
| | | RP | 6.78 | 31.07 | 73.73 | 19.49 | 9.32 | 14.69 | 0.73 |
| 10 | 2 | CCFM | **14.69** | 37.01 | **74.86** | **67.51** | **31.36** | **51.98** | 4.34 |
| | | FlexDock | 12.99 | **38.14** | 73.16 | 43.50 | 21.19 | 33.05 | 4.20 |
| | | RP | 13.56 | 36.72 | 73.16 | 44.07 | 21.19 | 34.18 | 4.39 |
| 20 | 2 | CCFM | 12.43 | **39.27** | **74.01** | 67.80 | 33.62 | 51.41 | 8.51 |
| | | FlexDock | **12.71** | **39.27** | 72.88 | 44.63 | 23.16 | 33.05 | 7.82 |
| | | RP | 11.58 | 37.85 | 72.88 | 49.72 | 25.42 | 37.85 | 9.21 |
| 1 | 10 | CCFM | **7.91** | **32.20** | 72.32 | **78.81** | 30.79 | **58.19** | 0.79 |
| | | FlexDock | 7.63 | 31.64 | **72.88** | 57.91 | 24.29 | 40.68 | 0.57 |
| | | RP | **7.91** | 31.92 | **72.88** | 56.21 | 27.40 | 42.66 | 0.92 |
| 1 | 20 | CCFM | **8.19** | **31.92** | 72.32 | **76.84** | 29.94 | **57.06** | 0.88 |
| | | FlexDock | 7.63 | 31.07 | 72.60 | 60.73 | 25.99 | 43.50 | 0.62 |
| | | RP | 7.91 | 31.36 | **72.88** | 50.85 | 26.55 | 40.68 | 1.10 |
| 10 | 10 | CCFM | **15.54** | 37.85 | 72.03 | **85.03** | **35.88** | **62.15** | 4.57 |
| | | FlexDock | **15.54** | **38.14** | 72.60 | 80.51 | 35.31 | 57.06 | 3.99 |
| | | RP | 13.84 | **38.14** | **72.88** | 76.55 | 34.75 | 56.78 | 5.15 |

outperforms the state-of-the-art baseline *FlexDock*, *with particularly large gains when the computational budget is limited*. Remarkably, with only one sample and two inference steps, CCFM achieves comparable L-RMSD and A-RMSD values, confirming that high-fidelity pose generation is preserved. At the same time, CCFM raises PB Valid from $20.34\%$ (FlexDock) to $47.74\%$, representing a **2.3**× improvement over FlexDock. With this setting, CCFM even surpasses FlexDock with twenty samples, yielding a **20**× improvement in sample–step efficiency (measured as samples × steps) and a **12**× gain in runtime (7.82s vs. 0.64s). On stricter metrics that jointly account for fidelity and feasibility (PB Valid+L and PB Valid+A), CCFM also achieves more than 2× higher pass rates, again matching the performance of FlexDock at 20 samples while using far fewer samples and computational time. Increasing the inference steps to 10 or 20 further amplifies this advantage: at 10 steps, CCFM improves PB Valid by $20.9\%$, PB Valid+L by $6.5\%$, and PB Valid+A by $17.5\%$, while simultaneously yielding higher fidelity and feasibility than the baseline. Comparable trends are observed when both the sample count and inference steps are set to ten. Additional comparisons across more metrics and experimental settings are provided in Appendix C.1.

Figure 2 compares the binding structures generated by our method (CCFM) and the FlexDock on the same protein–ligand pair. In molecular docking, protein–ligand atoms must stay at least the sum of their van der Waals radii minus 0.75 Å Corso et al. (2025). In this case, the van der Waals radii of the carbon (1.70 Å) and oxygen (1.52 Å) atoms yield a lower bound of 2.47 Å. As shown in Figure 2a, CCFM produces a feasible structure with the distance of 3.8 Å that satisfies the constraint, whereas FlexDock in Figure 2b generates an infeasible structure with a distance of 2.4 Å.

## 5.3 PERFORMANCE EVALUATION FOR PDE

We next focus on generating PDE solutions. Table 2 summarizes the performance of the different models tested. For the Reaction–Diffusion, the unconstrained *FFM* baseline yields relatively poor accuracy (MMSE $5.3 \times 10^{-2}$), while DiffusionPDE improves marginally (to $5.2 \times 10^{-2}$) by incorporating penalty terms during sampling that steer samples towards the prescribed initial condition and enforce the conservation law. However, this approach fails to handle non-linear constraints and does not guarantee feasibility. *PCFM* achieves a substantial accuracy gain (MMSE reduced to $3.6 \times 10^{-2}$) while simultaneously ensuring minimal constraint violations. Nevertheless, this procedure incurs a computational burden (see Appendix C) due to the projections at every sampling step, resulting in

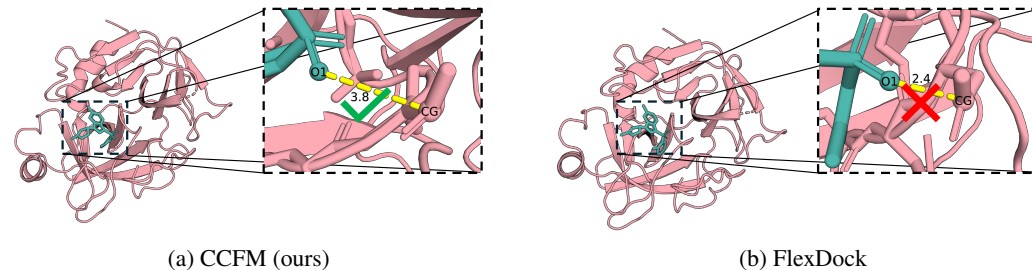

(a) CCFM (ours)                                    (b) FlexDock

Figure 2: Structures generated by CCFM (ours) and FlexDock. CCFM enforces the ligand–protein distance constraint, while FlexDock violates it.

| Problem | Metric | CCFM (ours) | PCFM | DiffusionPDE | FFM | RP |
|---|---|---|---|---|---|---|
| Reaction–Diffusion | MMSE $\times 10^{-2}$ ($\downarrow$) | **3.3** | 3.6 | 5.2 | 5.3 | 8.1 |
| | SMSE $\times 10^{-2}$ ($\downarrow$) | **2.7** | 2.9 | 3.8 | 3.8 | 8.5 |
| | CV (IC) $\times 10^{-2}$ ($\downarrow$) | **0** | **0** | 8.0 | 8.2 | **0** |
| | CV (CL) $\times 10^{-15}$ ($\downarrow$) | **9.5** | **9.5** | $3.9 \times 10^{10}$ | $3.9 \times 10^{10}$ | 10.8 |
| Navier–Stokes | MMSE $\times 10^{-1}$ ($\downarrow$) | **0.8** | 1.7 | 1.4 | 1.5 | 1.0 |
| | SMSE $\times 10^{-1}$ ($\downarrow$) | **0.8** | 1.6 | 8.8 | 9.0 | 0.9 |
| | CV (IC) ($\downarrow$) | **0** | **0** | 0.12 | 0.13 | **0** |
| | CV (CL) $\times 10^{-15}$ ($\downarrow$) | 3.9 | **3.4** | $1.2 \times 10^{10}$ | $1.2 \times 10^{9}$ | 4.1 |

Table 2: Comparison of model performance in solution generation tasks for 1D Reaction–Diffusion equation and 2D Navier–Stokes equation across metrics. A lower value indicates better performance. Bold indicates the best (smallest) value in each row.

a runtime that is approximately $28\times$ higher than the unconstrained FFM baseline *CCFM* further advances performance (MMSE $3.3 \times 10^{-2}$, SMSE $2.7 \times 10^{-2}$) while providing **zero constraint violations**. Moreover, *CCFM* achieves about a 30% faster runtime than PCFM (see Appendix C). Figure 3 further illustrates its MMSE across physical time, where the prevalence of darker regions indicates improved temporal stability.

On the Navier–Stokes task, the increase in dimension and the setting of the distribution test exacerbate the limitations of the baseline methods. *PCFM* yields higher MMSE and SMSE than DiffusionPDE despite preserving feasibility. Its reliance on one-step predictions from intermediate states leads to inaccuracies in high dimensions, and projection–interpolation steps amplify these errors, reducing accuracy. In contrast, *CCFM* avoids this failure mode by introducing only minimal corrective adjustments toward the feasible region at each step, without dependence on unstable one-step extrapolations. Consequently, CCFM achieves the best overall performance (MMSE $8.4 \times 10^{-2}$, SMSE $8.3 \times 10^{-2}$), while simultaneously ensuring feasibility. Figure 4 visualizes these results, showing that *CCFM* consistently suppresses error accumulation and aliasing under nonlinear dynamics.

## 5.4 PERFORMANCE EVALUATION FOR PHYSICS-INFORMED MOTION

Table 3 show that CCFM achieves the lowest constraint violation and slightly improved FID compared to the repeated projection approach. The superior FID of the unconstrained model, despite its large constraint violation, comes from the absence of background artifacts. Although projection and CCFM do not directly modify the background, the base model, which was trained under Earth gravity, tends to generate frames consistent with its training distribution. After each projection step moves the object to the Moon-gravity location, the model may partially pull it back toward its

Table 3: Comparison of model performance in the physics-informed motion task.

| | Unconstrained | Repeated Projection | CCFM |
|---|---|---|---|
| Constraint Violation (MSE) ($\downarrow$) | 278.46 | 0.8333 | **0.0100** |
| FID ($\downarrow$) | 16.985 | 27.184 | **26.767** |

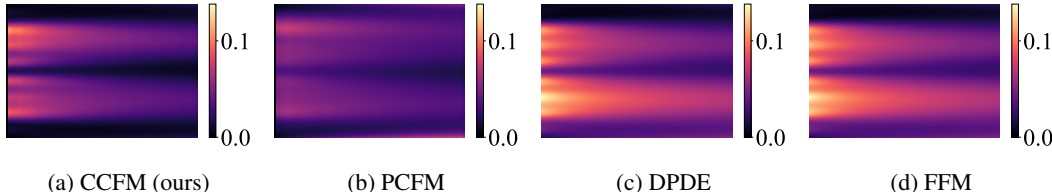

(a) CCFM (ours)    (b) PCFM    (c) DPDE    (d) FFM

Figure 3: MMSE (the darker the better) of Reaction–Diffusion solutions over physical time, with the horizontal axis representing time evolution and the vertical axis indicating the state at each time step.

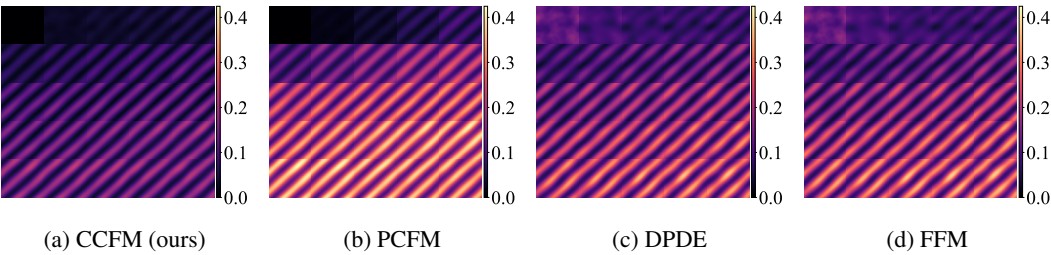

(a) CCFM (ours)    (b) PCFM    (c) DPDE    (d) FFM

Figure 4: MMSE (the darker the better) of Navier–Stokes solutions over physical time, downsampled from 50 to 25 frames for clarity. Each frame corresponds to the system state at a given physical time.

original in-distribution trajectory, creating slight ghosting in the background. In contrast, the unconstrained model leaves the background clean, even though the generated trajectories are far from physically feasible under Moon gravity. Therefore, for the constrained generation task where satisfying the physical dynamics is the primary objective, CCFM provides the most accurate and reliable results.

## 6 CONCLUSION

This paper introduced *Chance-constrained Flow Matching* (CCFM), a training-free framework for constraint-aware generative modeling. Its key idea is a reformulation of the repeated projection *during flow-matching sampling* through a stochastic-optimization lens. This enables the effective enforcement of hard constraints without introducing multi-stage procedures or additional train-time complexity. Our analysis established feasibility guarantees and, under the flow-linearity assumption used in our sampler, showed equivalence to projecting onto the clean feasible set, thus providing guarantees on constraint satisfaciton. The empirical evaluation against current state-of-the-art on molecular docking and PDE generation showed that CCFM attains strong gains, with especially large improvements under tight budgets (e.g., single-sample, few-step regimes) and markedly higher sample–step efficiency.

**Limitations and future work.** Despite the strong potential of CCFM, several directions remain open. In domains such as robotics and other scientific simulation applications, constraints may not admit closed forms. This aspect is out of the scope of the work, and the challenge of embedding non-differentiable black-box simulators (e.g., collision checkers, invariants) remains an interesting direction for future work.

## ETHICS STATEMENT

This work develops methods for constrained flow matching in scientific domains. Our method improves feasibility in both PDE solution generation and molecular docking, which holds significant potential to accelerate existing scientific discovery pipelines. To mitigate risks of potential misuse, this paper restricts evaluation to safe, publicly available structural benchmarks. All implementations follow standard open science practices, ensuring safe and transparent research

## REPRODUCIBILITY STATEMENT

The code, dataset, and checkpoints are released in the supplementary material along with instructions to guide the reproduction of the results presented in this paper. Methodological details are described extensively in the paper and accompanying appendix. Section 5, Appendix B, Appendix C describe the specific formulation, hyperparameters, models, and hardware used for evaluation pipelines.

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

## A    MISSING PROOF

### A.1    PROOF OF PROPOSITION 1

*Proof of Proposition 1.* Throughout, let $z_{1-\alpha} := \Phi^{-1}(1-\alpha)$ and $z_{1-\alpha/2} := \Phi^{-1}(1-\alpha/2)$. Write $\sigma_a := \sigma\|a\|_2$ and $\mu := t^{-1}a^\top x_t$. Note that $a^\top\xi \sim \mathcal{N}(0, \sigma_a^2)$.

**Linear constraints.**    Consider $g(x) = a^\top x - b$. The chance constraint

$$\mathbb{P}_\xi\big(a^\top(t^{-1}x_t - \xi) - b \le 0\big) \ge 1 - \alpha$$

is equivalent to

$$\mathbb{P}\big(a^\top\xi \ge \mu - b\big) \ge 1 - \alpha.$$

Since $a^\top\xi \sim \mathcal{N}(0, \sigma_a^2)$, the quantile characterization gives

$$\mu - b \le -\sigma_a z_{1-\alpha}.$$

and

$$a^\top x_t \le tb - t\sigma_a z_{1-\alpha} = tb - t\sigma\|a\|_2 z_{1-\alpha}.$$

**Quadratic constraints.**    Consider $g(x) = (a^\top x)^2 - b$ with $b > 0$. The chance constraint

$$\mathbb{P}_\xi\big((a^\top(t^{-1}x_t - \xi))^2 \le b\big) \ge 1 - \alpha$$

can be written as $\mathbb{P}(|S| \le \sqrt{b}) \ge 1 - \alpha$, where $S := a^\top(t^{-1}x_t - \xi) = \mu - a^\top\xi \sim \mathcal{N}(\mu, \sigma_a^2)$. By the union bound (Boole's inequality) Vershynin (2018),

$$\mathbb{P}(|S| \le \sqrt{b}) \ge 1 - \alpha \quad \Longleftarrow \quad \begin{cases} \mathbb{P}(S \le \sqrt{b}) \ge 1 - \frac{\alpha}{2}, \\ \mathbb{P}(S \ge -\sqrt{b}) \ge 1 - \frac{\alpha}{2}. \end{cases}$$

Using the Gaussian quantile characterization, these two one-sided constraints are respectively equivalent to

$$\sqrt{b} \ge \mu + \sigma_a z_{1-\alpha/2} \quad \text{and} \quad -\sqrt{b} \le \mu - \sigma_a z_{1-\alpha/2}.$$

Combining them yields

$$\sqrt{b} \ge \max\{\mu, -\mu\} + \sigma_a z_{1-\alpha/2} \quad \Longleftrightarrow \quad |\mu| \le \sqrt{b} - \sigma_a z_{1-\alpha/2}.$$

Equivalently, in terms of $x_t$,

$$|a^\top x_t| \le t\Big(\sqrt{b} - \sigma\|a\|_2 z_{1-\alpha/2}\Big).$$

For non-vacuous feasibility, it is necessary that $\sqrt{b} \ge \sigma_a z_{1-\alpha/2}$. Under these conditions, the original two-sided chance constraint is conservatively enforced with risk split $\alpha/2$ on each tail; the bound is tight when $\mu = 0$ (i.e., $a^\top x_t = 0$), since then

$$\mathbb{P}\big(|S| \le \sqrt{b}\big) = \mathbb{P}\big(|\mathcal{N}(0, \sigma_a^2)| \le \sqrt{b}\big) = 1 - 2\big(1 - \Phi(\tfrac{\sqrt{b}}{\sigma_a})\big),$$

and the condition $\sqrt{b} \ge \sigma_a z_{1-\alpha/2}$ is necessary and sufficient for $\mathbb{P}(|S| \le \sqrt{b}) \ge 1 - \alpha$.

Finally, for anisotropic noise $\xi \sim \mathcal{N}(0, \Sigma)$, replace $\sigma\|a\|_2$ by $\sqrt{a^\top\Sigma a}$ throughout; the above derivations remain unchanged since $a^\top\xi \sim \mathcal{N}(0, a^\top\Sigma a)$. $\square$

### A.2    PROOF OF COROLLARY 2

*Proof.* From Equation 10, we have $\xi = (1-t)t^{-1}x_0$. At $t = 1$, it follows that $\xi = 0$. Hence, for any $x'$,

$$\mathbb{P}_\xi\big(g(t^{-1}x_t - \xi) \le 0\big) = \mathbb{P}_\xi\big(g(t^{-1}x_t) \le 0\big) = \begin{cases} 1, & g(t^{-1}x_t) \le 0, \\ 0, & \text{otherwise.} \end{cases}$$

Therefore, at $t = 1$, the chance-constrained feasible set coincides with the deterministic set $\mathcal{F}_{\det} = \{x' : g(t^{-1}x_t) \le 0\}$. Consequently, $\mathcal{P}_{cc}(x_1, 1 - \alpha)$ reduces to the Euclidean projection onto $\mathcal{F}_{\det}$, which completes the proof. $\square$

## A.3 PROOF OF THEOREM 3

*Proof.* Since $t^{-1}x_t = t^{-1}\big((1-t)x_0 + tx_1\big) = x_1 + \frac{1-t}{t}x_0$ and $\xi = \frac{1-t}{t}x_0$, we have the pathwise identity

$$t^{-1}x_t - \xi \;=\; x_1,$$

which holds for every realization of $x_0$. Applying $g$ to both sides yields $g(t^{-1}x_t - \xi) = g(x_1)$ almost surely, hence the probability is 1 or 0 depending on whether $g(x_1) \leq 0$. $\qquad\square$

## A.4 PROOF OF THEOREM 4

*Proof.* For any $x_t \in \mathbb{R}^d$,

$$x_t \in \mathcal{C}_t(x_0) \iff g\big(M_t(x_t)\big) \leq 0 \iff M_t(x_t) \in \mathcal{C}_1.$$

Thus $x_t \in \mathcal{C}_t(x_0)$ iff there exists $x_1 \in \mathcal{C}_1$ with $x_t = (1-t)x_0 + tx_1$, which yields $\mathcal{C}_t(x_0) = (1-t)x_0 + t\,\mathcal{C}_1$.

For the projection identity, substitute $x_t = (1-t)x_0 + tx_1$ with $x_1 \in \mathcal{C}_1$:

$$\min_{x_t \in \mathcal{C}_t(x_0)} \|x_t - x\|_2^2 = \min_{x_1 \in \mathcal{C}_1} \|(1-t)x_0 + tx_1 - x\|_2^2 = \min_{x_1 \in \mathcal{C}_1} t^2 \|x_1 - M_t(x)\|_2^2.$$

Since $\mathcal{C}_1$ is nonempty, closed, and convex, the Euclidean projection $\mathbb{P}_1(M_t(x))$ exists and is unique. Let $x'^* = \mathbb{P}_1(M_t(x))$. The unique minimizer in the original variable is

$$\mathbb{P}_t(x) = (1-t)x_0 + tx'^* = (1-t)x_0 + t\,\mathbb{P}_1\big(M_t(x)\big),$$

as claimed. $\qquad\square$

# B IMPLEMENTATION DETAILS

## B.1 MOLECULAR DOCKING FORMULATION

Given a receptor with atom coordinates $\mathcal{Y} \in \mathbb{R}^{N_{\mathrm{rec}} \times 3}$ and a ligand template conformer $\mathcal{X}_0 \in \mathbb{R}^{N_{\mathrm{lig}} \times 3}$, we parameterize a candidate pose by a rigid-body motion $(R, t) \in \mathrm{SO}(3) \times \mathbb{R}^3$ together with a vector of rotatable-bond torsions $\phi \in \mathbb{T}^K$. The posed ligand is expressed as

$$\mathcal{X}(R, t, \phi) \;=\; R\,T_{\mathrm{lig}}(\mathcal{X}_0; \phi) \;+\; \mathbf{1}\,\Delta^\top,$$

where $T_{\mathrm{lig}}(\cdot; \phi)$ applies torsional rotations about designated bond axes. The goal of molecular docking is to recover ligand poses that faithfully reproduce experimentally observed binding modes while maintaining chemical validity and physical plausibility. This work considers four classes of constraints:

**Bond Lengths.** For each ligand bond $(i, j) \in \mathcal{E}_{\mathrm{bond}}$,

$$(1 - \delta_b)\,\ell_{ij}^{\mathrm{low}} \;\leq\; \|\mathcal{X}_i - \mathcal{X}_j\| \;\leq\; (1 + \delta_b)\,\ell_{ij}^{\mathrm{up}},$$

where $\mathcal{E}_{\mathrm{bond}}$ is the set of ligand covalent bonds, $\delta_b = 0.25$ denotes the tolerance, $\ell_{ij}^{\mathrm{low}}, \ell_{ij}^{\mathrm{up}}$ are chemically derived lower/upper bounds (in Å) for bond $(i, j)$, $\mathcal{X}_i \in \mathbb{R}^3$ and $\mathcal{X}_j \in \mathbb{R}^3$ denote the Cartesian coordinates (in Å) of the $i$-th and $j$-th ligand atom, respectively.

**Bond Angles.** For every bond-angle triplet $(i-j-k) \in \mathcal{E}_{\angle}$,

$$(1 - \delta_\alpha)\,\alpha_{ijk}^{\mathrm{low}} \;\leq\; \angle(i-j-k) \;\leq\; (1 + \delta_\alpha)\,\alpha_{ijk}^{\mathrm{up}},$$

where $\mathcal{E}_{\angle}$ denotes the set of bond-angle triplets, $\angle(i-j-k)$ is the geometric angle at vertex $j$, $\alpha_{ijk}^{\mathrm{low}}, \alpha_{ijk}^{\mathrm{up}}$ are lower/upper bounds (in radians) for angle $(i-j-k)$, $\delta_\alpha = 0.25$ denotes the bond angles tolerance.

**Internal Steric Clash (Intra-ligand).**  For all ligand atom pairs $(i, j)$ not directly bonded or forming an angle, i.e., $(i, j) \notin \mathcal{E}_{\mathrm{bond}} \cup \mathcal{E}_{\angle}$,

$$\|\mathcal{X}_i - \mathcal{X}_j\| \geq (1 - \delta_{\mathrm{steric}})\, d_{ij}^{\mathrm{low}},$$

where $d_{ij}^{\mathrm{low}}$ is the lower bound (in Å) on permissible separation for nonbonded, nonangle ligand pairs, $\delta_{\mathrm{steric}} = 0.2$ is the internal steric clash tolerance.

**Minimum Protein–Ligand Contact.**  Defining the closest cross-molecule distance as

$$d_{\min}(\mathcal{X}, \mathcal{Y}) = \min_{i \in \mathrm{lig},\, j \in \mathrm{rec}} \|\mathcal{X}_i - \mathcal{Y}_j\|,$$

We require that the van der Waals overlap does not exceed a tolerance $\delta_{\mathrm{pl}} = 0.75\text{Å}$:

$$\forall (i, j):\ d_{lp} \geq (\rho_i + \rho_j) - \delta_{\mathrm{pl}}, \quad \text{equivalently} \ \max_{i,j} \max\big(0,\, \rho_i + \rho_j - d_{ij}\big) < \delta_{\mathrm{pl}},$$

where $d_{ij}$ is the distance between ligand atom $i$ and protein atom $j$, and $\rho_i, \rho_j$ denote their van der Waals radii. Second, to ensure the ligand is not placed too far from the binding pocket, we require the existence of at least one contact:

## B.2    PDE Problem Formulation

### B.2.1    Reaction–Diffusion Equation

We consider the one-dimensional nonlinear reaction–diffusion equation

$$\frac{\partial v}{\partial t} = \nu \frac{\partial^2 v}{\partial s^2} + \rho\, v(1 - v),$$

where $v(s, t)$ denotes the state of the species at position $s$ and time $t$, $\nu > 0$ is the diffusion coefficient, and $\rho > 0$ is the reaction rate constant.

The solution $v(s, t)$ must satisfy the following two constraints:

**Initial Condition.**
$$-\delta_{\mathrm{pde}} \leq v(s, 0) - v_{\mathrm{IC}}(s) \leq \delta_{\mathrm{pde}},$$

where $v_{\mathrm{IC}}(s)$ denotes the initial state, $\delta_{\mathrm{pde}} = 1 \times 10^{-13}$ is the tolerance. This condition ensures that the solution starts from a specific initial condition $v_{\mathrm{IC}}(s)$ at time $t = 0$.

**Nonlinear Mass Conservation.**

$$-\delta_{\mathrm{pde}} \leq m(t) - \left( m(0) + \int_0^t \rho\nu(1 - \nu)dt + \int_0^t (g_L - g_R)\, dt \right) \leq \delta_{\mathrm{pde}},$$

where $m(t) = \int v(s, t)\, ds$ is the total mass at time $t$, and $g_L$, $g_R$ are the Neumann boundary fluxes at the left and right boundaries, respectively.

These constraints ensure consistency with the prescribed initial condition and the correct nonlinear evolution of the total mass due to reaction and boundary effects.

### B.2.2    Navier—Stokes Equation

The dynamics of incompressible fluids are governed by the Navier–Stokes equations:

$$\frac{\partial \mathbf{v}}{\partial t} + (\mathbf{v} \cdot \nabla)\mathbf{v} = -\nabla \pi + \nu \nabla^2 \mathbf{v} + \mathbf{f}, \qquad \nabla \cdot \mathbf{v} = 0,$$

where $\mathbf{v}(s, t)$ is the velocity field, $\pi(s, t)$ is the pressure field, $\nu > 0$ is the kinematic viscosity, $\mathbf{f}(s, t)$ is an external force (e.g., gravity).

The solution must satisfy the following two constraints:

**Initial Condition.**

$$-\delta_{\text{pde}} \ \le \ \mathbf{v}(s,0) - \mathbf{v}_{\text{IC}}(s) \ \le \ \delta_{\text{pde}},$$

where $\mathbf{v}_{\text{IC}}(s)$ is the initial state. This condition ensures the velocity field starts from a prescribed initial condition.

**Global Mass Conservation.**

$$-\delta_{\text{pde}} \ \le \ \iint \mathbf{v}(s,t)\,ds \ - \ \iint \mathbf{v}(s,0)\,ds, \ \le \ \delta_{\text{pde}},$$

ensuring that the total mass of the incompressible fluid remains constant over time.

### B.3 Physics-Informed Motion Formulation

We model the vertical motion of a rigid object falling under uniform gravity. Each trajectory contains six frames corresponding to a physical duration of $4$ seconds. The horizontal location is fixed, and only the vertical coordinate evolves according to the gravitational dynamics.

**Free-Fall Dynamics.** Let $\mathbf{p}_t \in \mathbb{R}^2$ denote the object position at frame $t$, and let $\mathbf{v}_t$ denote its velocity. The motion obeys the discrete-time update rule

$$\mathbf{p}_t = \mathbf{p}_{t-1} + \left( \mathbf{v}_t + 0.5\,\frac{\partial \mathbf{v}_t}{\partial t} \right), \tag{12a}$$

$$\mathbf{v}_{t+1} = \frac{\partial \mathbf{p}_t}{\partial t} + \frac{\partial \mathbf{v}_t}{\partial t}, \tag{12b}$$

where $\frac{\partial \mathbf{v}_t}{\partial t}$ is the gravitational acceleration. During training we use Earth gravity

$$\frac{\partial \mathbf{v}_t}{\partial t} \ = \ \mathbf{a}_{\text{E}} = (0, -9.8),$$

and during evaluation we impose Moon gravity

$$\frac{\partial \mathbf{v}_t}{\partial t} \ = \ \mathbf{a}_{\text{M}} = (0, -1.62).$$

We use $2048$ trajectories for training and $205$ for validation. For evaluation, we generate $100$ trajectories under Moon gravity using the three methods (unconstrained generation, repeated projection, and CCFM).

### B.4 Constraints Enforcement

#### B.4.1 Molecular Docking

To enforce geometric feasibility on molecular conformations, we adopt an iterative, gradient-free projection strategy. Starting from an initial state $x^{(0)}$, the algorithm generates a sequence of iterates $x^{(k+1)} = \text{Project}(x^{(k)})$, where each projection step reduces the violation of geometric constraints. The process terminates once no constraint is violated or a fixed iteration budget is reached, yielding a final feasible conformation.

At each step, we first identify the active set of violated constraints, denoted $\mathcal{A}(x)$. For each violated constraint, we compute a corrective displacement along the axis connecting the two atoms involved. The displacement magnitude is proportional to the violation and scaled by a small fixed step size, producing a push or pull effect that nudges atoms toward feasibility. The updates across all active constraints are aggregated into a single correction $\Delta x$, with per-atom displacements capped to ensure numerical stability. The coordinates are then updated as $x \leftarrow x + \Delta x$.

While a single update reduces constraint violations, it may not achieve strict feasibility. Iterative application guarantees convergence to a geometrically valid conformation. By restricting updates to the active set, the method avoids unnecessary corrections on already satisfied constraints, while still ensuring that all geometric conditions are satisfied at convergence.

### B.4.2 PDE

Following the enforcement strategy of Utkarsh et al. (2025), we also adopt the Gauss–Newton method for enforcing constraints in PDE formulations. For linear constraints, the projection is exact and guarantees feasibility. For nonlinear constraints, the single projection provides only an approximate correction, and strict feasibility is deferred to the final step, where multiple Gauss–Newton iterations are applied. In addition, since our setting considers inequality constraints, we adopt an *active set strategy*. Specifically, at each step we identify the subset of constraints that are currently violated and apply projection updates only to this active set. By restricting correction to infeasible constraints, this approach avoids unnecessary computations on already satisfied constraints, while still ensuring that all constraints are satisfied at convergence.

It is important to highlight that, although all constraint enforcement is based on the Gauss–Newton method, our approach differs from PCFM. In PCFM, projections are repeatedly applied to the original constraints defined on the clean sample, regardless of sampling time. In contrast, our method introduces a *chance-constrained projection operator*, which adaptively adjusts the tightness of constraints according to the sampling time. This adaptive mechanism is supported by theoretical guarantees, ensuring both efficiency and correctness.

**Linear Projection**  For linear inequality constraints, projection is exact. At each sampling step, we solve

$$x_{\mathrm{proj}} \;=\; \arg\min_z \|z - x\|_2^2 \quad \text{s.t.} \quad g(z) \le 0,$$

which admits a closed-form solution due to the rank-1 structure of the Jacobian. For instance, in the Navier—Stokes Equation, the total-vorticity constraint reduces to a constant-shift correction

$$\mathbf{v}(\cdot,\cdot,k) \;\leftarrow\; \mathbf{v}(\cdot,\cdot,k) - \text{average residual}.$$

This closed-form update yields exact feasibility at negligible cost and is applied after every sampling step.

**Gauss–Newton Projection**  For general nonlinear constraints, we employ an approximate correction. Given a candidate $x \in \mathbb{R}^n$ subject to $g_i(x) \le 0$, we first identify the active set of violated constraints

$$\mathcal{A}(x) = \{\, i \mid g_i(x) > 0 \,\},$$

with residuals and Jacobian

$$r = g_{\mathcal{A}(x)}(x), \qquad J = \nabla g_{\mathcal{A}(x)}(x)^\top \in \mathbb{R}^{n \times |\mathcal{A}(x)|}.$$

A single Gauss–Newton update

$$x_{\mathrm{proj}} = x - J^\top (JJ^\top)^{-1} r$$

reduces the residuals of the active constraints, pushing the iterate back toward the feasible region. In PCFM and CCFM, this step is applied after each sampling iteration.

Specifically, we implement the active set via hinge residuals $r = \mathrm{ReLU}(g(x))$ for efficient GPU batching. The linear system is stabilized by adding the regularization

$$y = (JJ^\top + \lambda I)^{-1} r, \quad \Delta x = -J^\top y, \quad x \leftarrow x + \Delta x,$$

with $\lambda \approx 10^{-6}$.

**Final Projection for Strict Feasibility**  To guarantee exact satisfaction of all constraints in the generated samples, we perform Newton–Schur iterations restricted to the active set:

$$x^{(k+1)} = x^{(k)} - J_k^\top (J_k J_k^\top)^{-1} g_{\mathcal{A}(x^{(k)})}\big(x^{(k)}\big), \quad J_k = \nabla g_{\mathcal{A}(x^{(k)})}(x^{(k)})^\top.$$

Under smoothness and full-rank assumptions, these iterations converge quadratically (Utkarsh et al., 2025). In both PCFM and CCFM, we allocate a fixed budget of 30 iterations after the final sampling step to ensure strict feasibility.

### B.5 DATASET DETAILS

#### B.5.1 MOLECULAR DOCKING

This work utilizes the same dataset as adopted in (Corso et al., 2025), namely the PDBBind benchmark (Liu et al., 2017), which has become a standard for evaluating docking models. Following previous work (Stärk et al., 2022; Corso et al., 2025), we employ the time-based split, where approximately 17,000 complexes released before 2019 are used for training and validation, while the 363 complexes released after 2019 constitute the test set.

To ensure consistency between apo and holo structures, we first sanitize the raw PDBBind files using PDBFixer (Eastman et al., 2017) to replace non-standard residues and add missing heavy atoms. For the apo state, we extract protein sequences from the processed files and generate structural predictions with ESMFold (Lin et al., 2022). These predictions are further refined by PDBFixer to add missing terminal atoms, while the crystallographic bound conformations from PDBBind serve as the holo state. Finally, hydrogen atoms are removed during alignment to reduce inconsistencies arising from variable hydrogen placement.

#### B.5.2 PDE

Following the dataset construction strategies in Cheng et al. (2024); Utkarsh et al. (2025), we generate training and testing trajectories by combining randomized initial conditions with varying boundary or forcing configurations. The key distinction is that, whereas prior works typically employ only 10 initial conditions in the test set, we adopt 90. This design probes model performance under more diverse and complex scenarios, thereby yielding a more challenging and discriminative evaluation.

**Reaction–Diffusion Equation** The 1D reaction–diffusion system is simulated on the spatial domain $s \in [0, 1]$ with $n_s = 128$ grid points and over $n_t = 100$ discrete time steps. The numerical simulations were performed using FiPy (Guyer et al., 2009), a Python-based finite volume solver for partial differential equations. We employ a semi-implicit scheme with a fixed diffusion coefficient and reaction rate of $(\nu, \rho) = (0.005, 0.01)$.

The initial conditions are drawn from randomized combinations of sinusoidal modes and localized perturbation functions, providing both smooth and sharp spatial profiles. Neumann boundary conditions are enforced with time-varying fluxes $(g_L, g_R)$ sampled from a prescribed range.

The dataset is split into training and testing sets. To form the training set, we pair 100 distinct initial conditions with 100 unique boundary flux configurations, yielding a total of 10,000 solution trajectories. For the test set, a separate pool of data is first generated by combining 90 new initial conditions with 90 new boundary flux configurations; from this pool of 8,100 trajectories, 1,000 are then randomly selected for evaluation. The conditions used for the training and test sets are entirely disjoint.

**Navier—Stokes Equation** We consider the 2D incompressible Navier–Stokes equations in vorticity form on a periodic square domain discretized with a $64 \times 64$ spatial grid. Temporal evolution is simulated over $T = 49$ time units, with 50 uniformly sampled snapshots recorded per trajectory. We adopt a Crank–Nicolson spectral solver to ensure stability for the viscosity parameter $\nu = 10^{-3}$. The initial vorticity fields are sampled from Gaussian random fields with prescribed covariance structure, while the external forcing is defined as

$$f(x) = \frac{0.1}{\sqrt{2}} \sin\big(2\pi(x_1 + x_2) + \varphi\big), \qquad \varphi \sim \mathcal{U}(0, \pi/2).$$

To construct the training dataset, we generate 10,000 trajectories by sampling 100 random initial conditions and 100 random forcing phases. For evaluation, a separate dataset is first generated using 90 new initial conditions and 90 new forcing phases. From this pool of 8,100 trajectories, we then randomly select 1,000 trajectories to form the final test set. The initial conditions used for training and evaluation are disjoint.

## B.6 MODEL DETAILS

### B.6.1 MOLECULAR DOCKING

For our experiments, we adopt the FlexDock framework, a state-of-the-art generative model for flexible molecular docking based on Unbalanced Flow Matching (UFM) (Corso et al., 2025). FlexDock integrates two complementary modules: a manifold docking flow, which operates in reduced degrees of freedom to efficiently capture ligand rigid-body motions and protein side-chain flexibility, and a subsequent relaxation flow, which refines atomic coordinates in Euclidean space under energy-based constraints to ensure physically valid conformations.

This work directly utilizes the official pre-trained FlexDock models provided by the authors, trained on the PDBBind dataset with time-based splits. This setup allows us to leverage a robust, well-validated backbone without additional fine-tuning, thereby ensuring reproducibility, comparability with prior benchmarks, and fairness in experimental comparisons.

## B.7 PDE

For all PDE benchmarks, we adopt the Functional Flow Matching (FFM) framework parameterized by a Fourier Neural Operator (FNO) backbone (Kerrigan et al., 2023). The FNO encoder processes the concatenation of the current state, a sinusoidal positional embedding of the spatial grid, and a Fourier time embedding, thereby enabling spatiotemporal generalization across functional domains. The resulting time-dependent vector field defines the flow dynamics used in all experiments. Table 4 and Table 5 summarize the hyperparameters used during training for reaction–Diffusion and Navier–Stokes equations, respectively.

Table 4: Params for Reaction–Diffusion.

| Hyperparameter | Value |
| --- | --- |
| Fourier Layers | 6 |
| Fourier Modes | [32,32] |
| Hidden Channels | 64 |
| Projection Channels | 256 |
| Batch Size | 256 |
| Optimizer | adam |
| Scheduler | plateau |
| Prior Distribution | randn |
| Training Steps | 20000 |

Table 5: Params for Navier–Stokes.

| Hyperparameter | Value |
| --- | --- |
| Fourier Layers | 4 |
| Fourier Modes | [16,16,16] |
| Hidden Channels | 64 |
| Projection Channels | 256 |
| Batch Size | 16 |
| Optimizer | adamw |
| Scheduler | plateau |
| Prior Distribution | randn |
| Training Steps | 200000 |

## B.8 SAMPLING SETUP

### B.8.1 MOLECULAR DOCKING

FlexDock employs two distinct flow-matching models for molecular docking: one dedicated to docking and the other to relaxation. Constraint enforcement is primarily handled by the second model, which applies an Euler-based update scheme. At each inference step, multiple candidate samples are generated, and a discriminator is subsequently used to select the single most promising configuration as the final output. We evaluate FlexDock under varying sample counts (1, 5, 10, 15, 20) and inference steps (2, 5, 10, 15, 20), reporting the detailed results in Section 5 and the remaining results in the appendix C.

CCFM adopts a sampling setup largely consistent with FlexDock, employing the same ranges of sample counts and inference steps. The only distinction is that CCFM enforces constraints after each inference step using a gradient-free method shown in Appendix B.4, followed by a final refinement of 20 iterations once the sampling trajectory is complete.

Table 6: Sensitivity analysis for probability of constraint satisfaction. The probability is determined by the function $\phi(t) = (t/2)^n$, and we adjust $n$ to observe the results. Results are reported using #S=1 and #Stp=10. PB Valid (+L < 2Å or +A < 2Å) denotes the percentage of physically valid samples that also satisfy the respective RMSD thresholds.

| Metric | CCFM(0.7) | CCFM(0.9) | CCFM(1.0) | CCFM(1.1) | CCFM(1.3) |
|---|---|---|---|---|---|
| PB Valid (+L < 2Å) | 29.85 | 30.16 | 30.79 | 30.48 | 30.48 |
| PB Valid (+A < 2Å) | 58.19 | 57.85 | 58.19 | 58.19 | 58.88 |

Table 7: PDBBind docking results with standard deviation (#S=1, #Stp=10).

| Methods | L-RMSD (%) | | A-RMSD (%) | PB Valid (%) | | |
|---|---|---|---|---|---|---|
| | < 1Å | < 2Å | < 2Å | All | +L < 2Å | +A < 2Å |
| CCFM | $7.80 \pm (0.31)$ | $32.57 \pm (0.57)$ | $72.27 \pm (0.69)$ | $79.15 \pm (0.41)$ | $30.45 \pm (0.68)$ | $58.53 \pm (0.49)$ |
| FlexDock | $7.45 \pm (0.43)$ | $31.92 \pm (0.81)$ | $72.65 \pm (0.81)$ | $58.07 \pm (1.24)$ | $24.57 \pm (1.13)$ | $40.96 \pm (1.38)$ |
| RP | $7.80 \pm (0.34)$ | $32.19 \pm (0.79)$ | $72.26 \pm (0.99)$ | $56.48 \pm (0.72)$ | $27.68 \pm (0.81)$ | $42.93 \pm (0.85)$ |

### B.8.2 PDE

We adopt high-order ODE solvers (Dopri5) for unconstrained generation. For all constraint-aware methods, we employ the modified Euler scheme (Heun's method) with 200 sampling steps to ensure a fair comparison.

**CCFM (ours).** Our method performs explicit Euler integration with constraint correction at every step. Each intermediate state is projected onto the constraint manifold using a single Newton update. As shown in Proposition 1, the deterministic reformulation of chance constraints is governed by both the variance of the induced random variable and the probability of constraint satisfaction. Specifically, the variance depends on the sampling time $t$, while the satisfaction probability is given by $\phi(t) = (t/2)^n$. The parameter $n$ thus controls the tightening schedule: larger values (e.g., $n = 0.5$) enforce stricter constraints at later stages, while smaller values (e.g., $n = 0.1$) apply stronger corrections earlier. We adopt $n = 0.5$ for the reaction–diffusion task and $n = 0.1$ for the Navier–Stokes task. A detailed sensitivity analysis with respect to $n$ is provided in the Appendix C. After the sampling process, we apply 30 iterations of Gauss–Newton refinement to guarantee the strict feasibility of all constraints.

**PCFM.** This baseline relies on iterative extrapolation–correction–interpolation steps combined with periodic mixing and noise resampling. To align with our fixed sampling budget, we restrict the number of mixing steps to 2. Constraint correction follows the same projection-based strategy as in CCFM.

**Diffusion PDE.** We follow the gradient-guided sampling procedure (Huang et al., 2024) on a pretrained FFM backbone. Constraint satisfaction is encouraged by augmenting the generation loss with explicit penalties on both the initial condition and the conservation law. The composite loss is weighted by a factor of 200 to balance fidelity and feasibility.

### B.9 HARDWARE

For the molecular docking experiments, inference was conducted on six NVIDIA A6000 GPUs. For the Reaction–Diffusion, both training and inference were performed on a single NVIDIA A100 GPU. For the Navier–Stokes, the training process was done on six NVIDIA A6000 GPUs, while inference was performed on a single NVIDIA A100 GPU.

## C ADDITIONAL RESULTS

### C.1 MOLECULAR DOCKING

A comparison between CCFM and FlexDock across varying sample sizes and inference steps is shown in Figure 5.

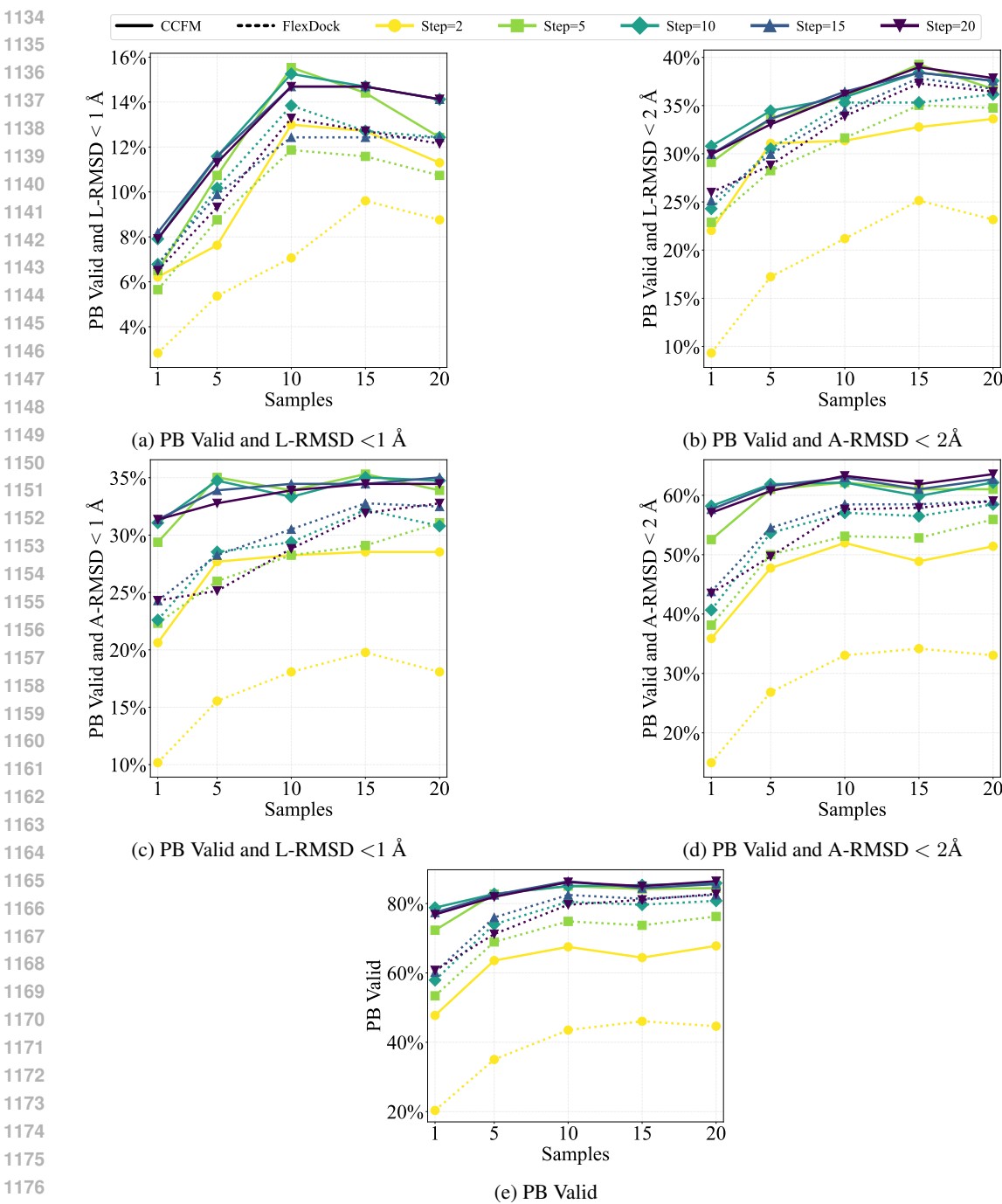

Figure 5: Comparison of CCFM (solid lines) and FlexDock (dashed lines) under different sampling budgets and evaluation metrics.

At the strictest setting (PB Valid with L-RMSD < 1 Å, step = 2, sample = 1), CCFM attains over 6% success rate, more than doubling FlexDock's 3%. A similar advantage persists when relaxing the cutoff to L-RMSD < 2 Å, where CCFM consistently surpasses FlexDock across all sample and step budgets.

When evaluated on A-RMSD, the margin becomes even more pronounced: for PB Valid with A-RMSD < 1 Å, CCFM exceeds FlexDock by at least 5% in nearly every configuration, and maintains comparable superiority at the A-RMSD < 2 Å threshold.

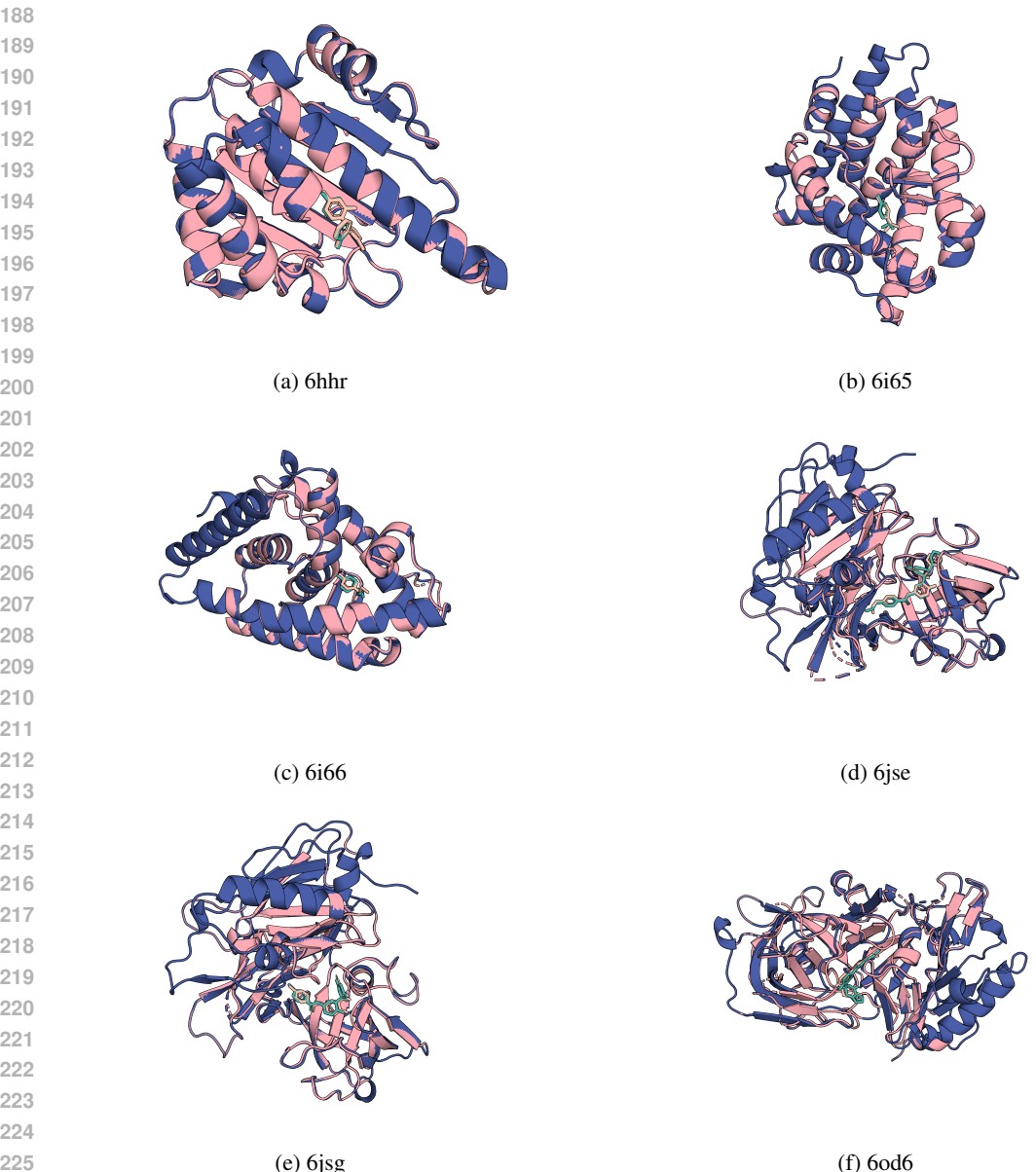

(a) 6hhr

(b) 6i65

(c) 6i66

(d) 6jse

(e) 6jsg

(f) 6od6

Figure 6: Visualization for the generated complexes(6HHR, 6I65, 6I66, 6JSE, 6JSG, and 6OD6) from the PDBBind test dataset. Predicted protein structures are shown in pink, predicted ligands in teal, ground-truth proteins in dark blue, and ground-truth ligands in melon.

For PB Valid, with a moderate budget (5 samples, 5 steps), CCFM achieves a PB Valid level comparable to FlexDock with 20 samples and 20 steps, reducing the effective sample-step cost **from 400 to just 25**. These results highlight the overall trend: CCFM consistently matches or exceeds FlexDock validity within a fraction of the computational expense.

The improvements are most striking under limited budgets, where CCFM achieves strong validity and fidelity with far fewer samples and steps. Even as the computational budget increases, CCFM continues to deliver higher feasibility and fidelity while maintaining a clear advantage in efficiency.

Figure 6 illustrates representative docking results on six PDBBind test complexes, highlighting the alignment between predicted and ground-truth protein–ligand structures.

Table 8: Comparison of running time between CCFM and PCFM on Reaction–Diffusion and Navier–Stokes equations.

| Problem | Metric | CCFM (ours) | PCFM |
|---|---|---|---|
| Reaction–Diffusion | Running Time (s) ($\downarrow$) | 2.703 | 3.874 |
| Navier–Stokes | Running Time (s) ($\downarrow$) | 3.065 | 3.077 |

Table 9: Sensitivity analysis for probability of constraint satisfaction. The probability is determined by the function $\phi(t) = (t/2)^n$, and we adjust $n$ to observe the results.

| Problem | Metric | CCFM (0.1) | CCFM (0.3) | CCFM (0.5) | CCFM (0.7) | CCFM (0.9) |
|---|---|---|---|---|---|---|
| Reaction–Diffusion | MMSE $\times 10^{-2}$ ($\downarrow$) | 3.6 | 3.6 | 3.3 | 3.2 | 3.2 |
| | SMSE $\times 10^{-2}$ ($\downarrow$) | 2.8 | 2.9 | 2.7 | 2.6 | 2.6 |
| | CV (IC) $\times 10^{-2}$ ($\downarrow$) | 0 | 0 | 0 | 0 | 0 |
| | CV (CL) $\times 10^{-15}$ ($\downarrow$) | 9.9 | 9.6 | 9.5 | 9.5 | 9.5 |

Table 6 shows that varying the probability scheduler parameter $n$ has a negligible impact on MMSE and SMSE while consistently maintaining constraint satisfaction. This demonstrates the method's robustness to the choice of scheduler.

Table 7 presents the mean $\pm$ standard deviation results from 5 independent experimental runs, validating docking methods on the PDBBind dataset. CCFM consistently demonstrates superior performance and stability compared to FlexDock and RP.

## C.2 PDE SOLUTION GENERATION

We report additional results of the PDE solution generation tasks. This section presents the running time for different methods, sensitivity analysis of the probability scheduler, MMSE curves of the generated solution fields varying with physical time, the visualization of SMSE and MMSE with all time frames (Navir–Stokes), the point-wise average residuals varying with physical time, and the comparison of models using an example of the generated solution field.

Table 8 shows that CCFM runs faster than PCFM while still ensuring feasibility. The gain is most pronounced in Reaction–Diffusion, where nonlinear projections slow down PCFM. By adaptively relaxing constraints with the chance-constrained operator, CCFM avoids unnecessary projections and reduces computational cost.

### C.2.1 REACTION–DIFFUSION

Table 9 presents the sensitivity analysis of the probability scheduler hyperparameter. We observe that varying the parameter $n$ has almost no impact on constraint satisfaction, consistently ensuring feasibility across all settings. The effects on MMSE and SMSE are also marginal, with only slight fluctuations that remain within a reasonable range. This indicates that the method is robust to the choice of the probability scheduler.

Figure 7 shows the MMSE curves as a function of physical time for the Reaction–Diffusion equation. CCFM demonstrates superior performance compared with the other models used in this study, while exactly satisfying the initial condition (with error equal to 0). PCFM has similar performance. Figure 8 shows the SMSE, where the x-axis represents physical time and the y-axis represents space. Similar to the MMSE (Figure 3), the proposed CCFM achieves the lowest errors compared with other models. The solution residuals, shown in Figure 9 demonstrates the improvement from CCFM compared with other models in generating a specific solution field. The CCFM-generated sample closely adheres to the ground truth, while all other models show substantial discrepancies. These additional results consistently highlight the superiority of the proposed CCFM in generating solutions with high accuracy while ensuring constraint satisfaction.

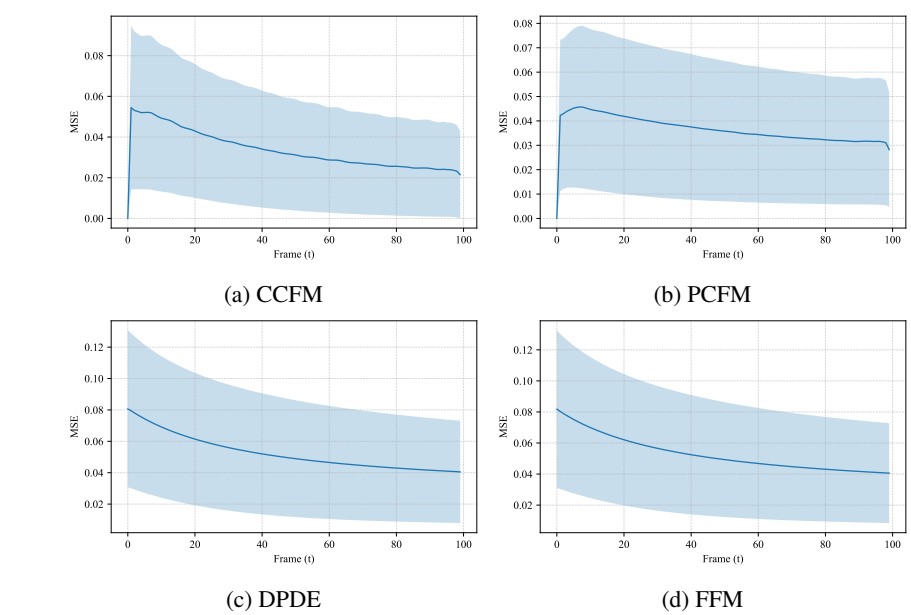

Figure 7: Comparison of generated solutions for the Reaction–Diffusion equation in terms of MMSE varying with physical time.

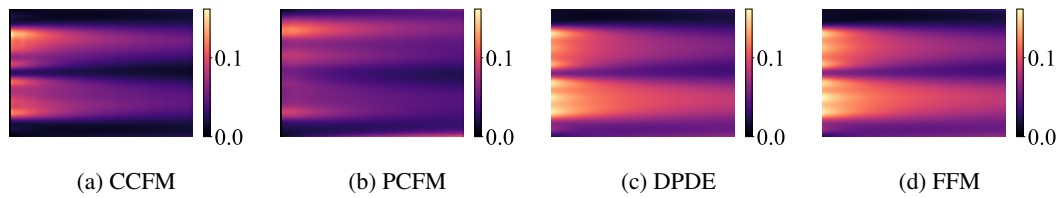

Figure 8: SMSE of Reaction–Diffusion solutions over physical time, with the horizontal axis representing time evolution and the vertical axis indicating the state at each time step.

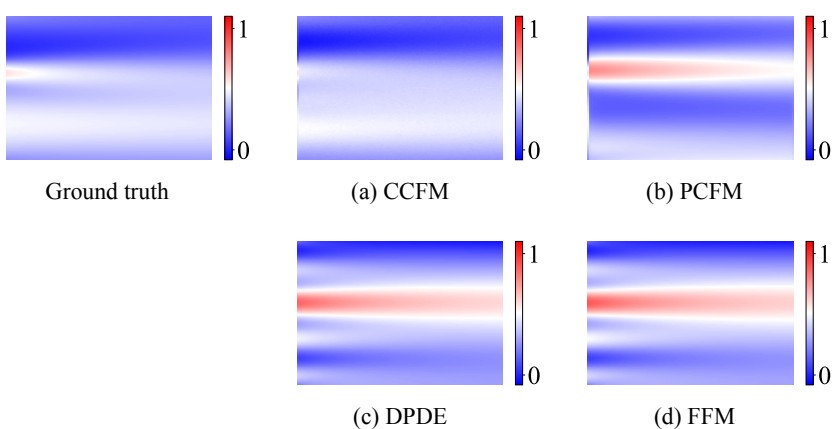

Figure 9: Example solution fields of the Reaction–Diffusion equation generated using the models compared with the ground truth.

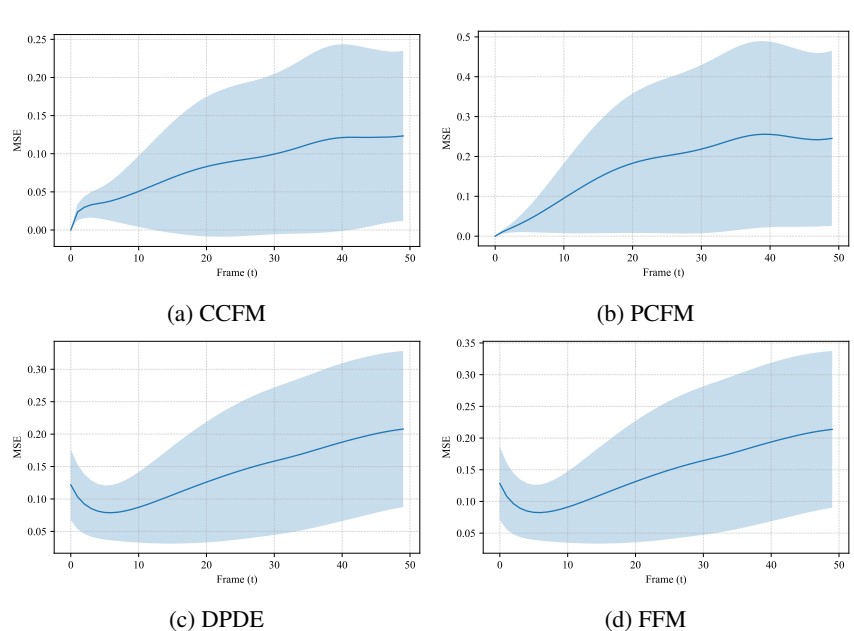

Figure 10: Comparison of generated solutions for the Navir–Stokes equation in terms of MMSE varying with physical time.

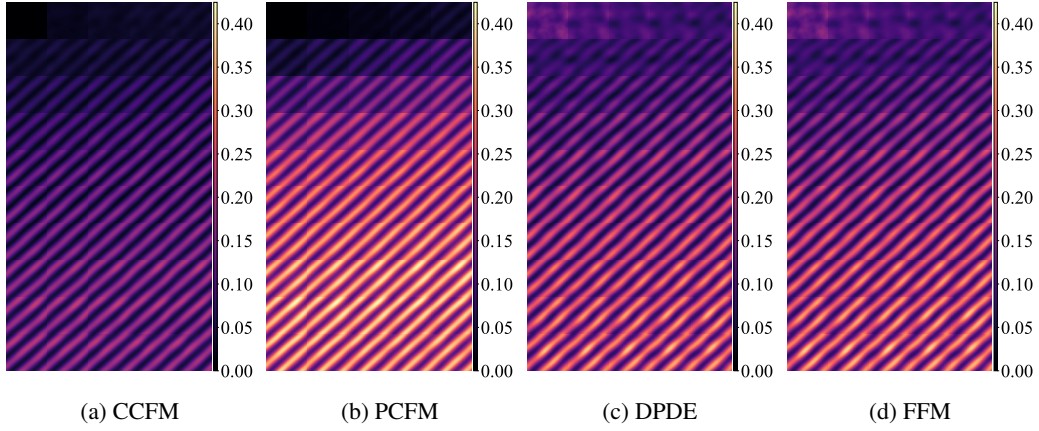

Figure 11: MMSE (the darker the better) of Navier–Stokes solutions over physical time with all 50 frames. Each frame corresponds to the system state at a given physical time.

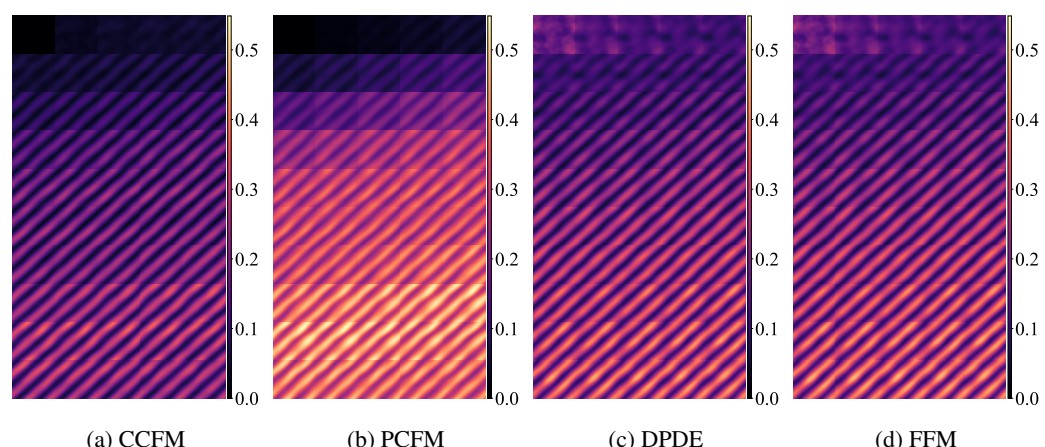

(a) CCFM      (b) PCFM      (c) DPDE      (d) FFM

Figure 12: SMSE (the darker the better) of Navier–Stokes solutions over physical time with all 50 frames. Each frame corresponds to the system state at a given physical time.

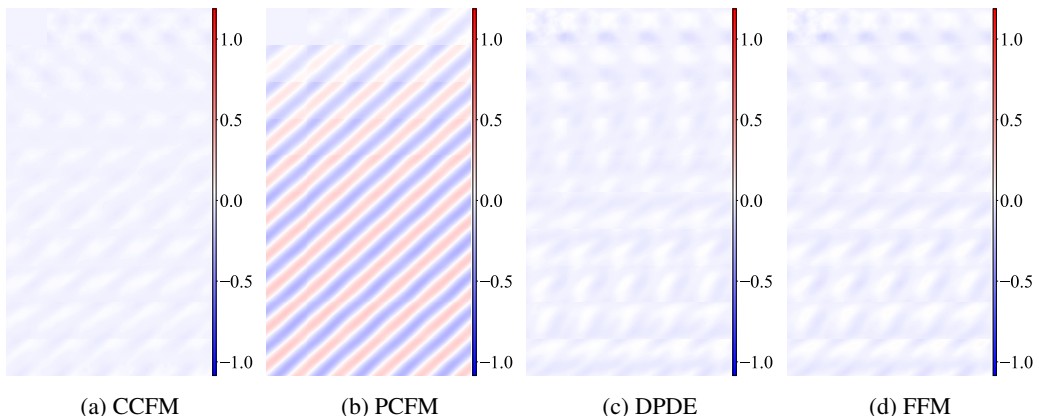

(a) CCFM      (b) PCFM      (c) DPDE      (d) FFM

Figure 13: Point-wise averaged residuals (the lighter the better) of Navier–Stokes solutions over physical time with all 50 frames. Each frame corresponds to the system state at a given physical time.

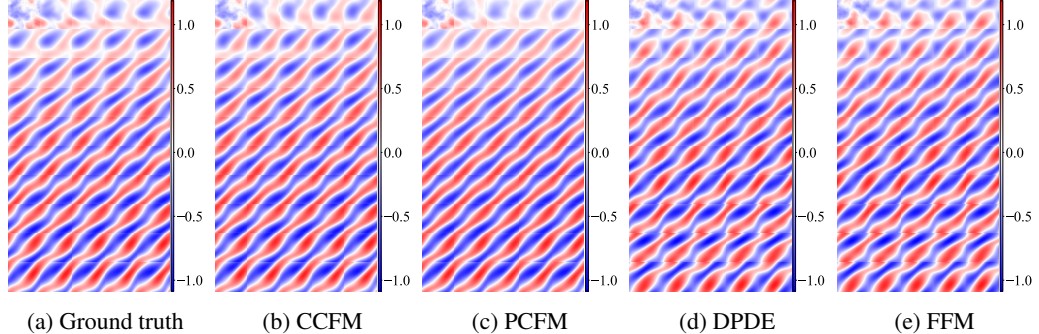

(a) Ground truth    (b) CCFM    (c) PCFM    (d) DPDE    (e) FFM

Figure 14: Example solution fields of the Navir–Stokes equation generated using the models compared with the ground truth.

Table 10: Results with standard deviations for solution generation on the 2D Navier–Stokes equation. A lower value indicates better performance. Bold indicates the best value in each row.

| Problem | Metric | CCFM (ours) | PCFM | DiffusionPDE | FFM | RP |
|---|---|---|---|---|---|---|
| Navier–Stokes | MMSE $\times 10^{-1}$ ($\downarrow$) | $0.79 \pm (0.03)$ | $1.67 \pm (0.09)$ | $1.44 \pm (0.10)$ | $1.56 \pm (0.16)$ | $1.08 \pm (0.08)$ |
| | SMSE $\times 10^{-1}$ ($\downarrow$) | $0.81 \pm (0.11)$ | $1.53 \pm (0.18)$ | $9.21 \pm (0.39)$ | $9.57 \pm (0.53)$ | $0.97 \pm (0.25)$ |
| | CV (IC) ($\downarrow$) | $0$ | $0$ | $0.12 \pm (0.03)$ | $0.13 \pm (0.08)$ | $0$ |
| | CV (CL) $\times 10^{-15}$ ($\downarrow$) | $3.9 \pm (0.01)$ | $3.4 \pm (0.02)$ | $1.2 \pm (0.13) \times 10^{10}$ | $1.2 \pm (0.12) \times 10^{9}$ | $4.1 \pm (0.04)$ |

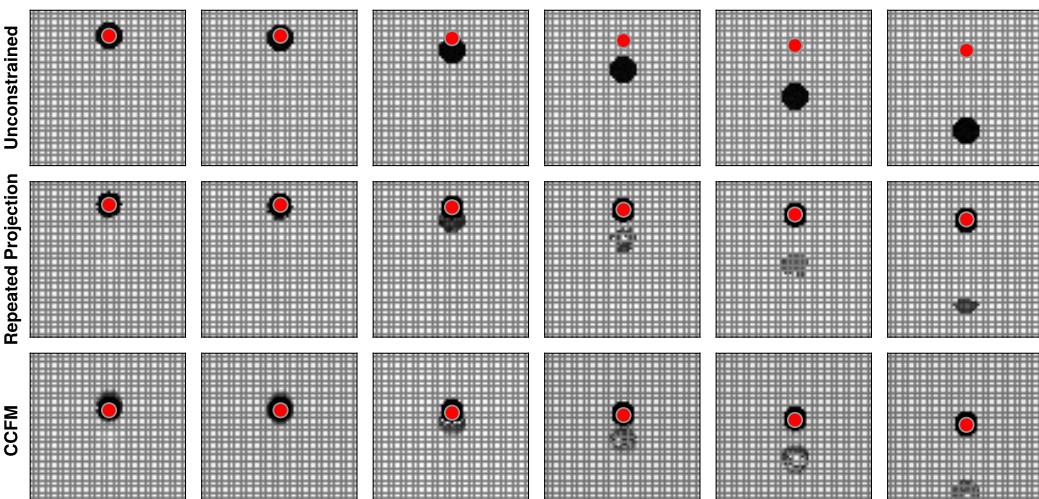

Figure 15: Generated trajectories instances for the physics-informed motion generation. In each frame, the black solid objects are generated using three constraint-handling approaches. The red circles represent the correct locations of objects under Moon gravity.

### C.2.2 NAVIER–STOKES

For the Navier–Stokes problem, to complement the results presented in Section 5.3, we report evaluation results for all physical time frames. Figure 11 shows the MMSE of generated samples from the models as a function of physical time. While the errors increase over time, CCFM maintains relatively low error levels across all frames compared with other models. This is also reflected in Figure 10, where CCFM outperforms other models, with the maximum spatially averaged MMSE around 0.12 (others > 0.20). Meanwhile, the average solution residuals from CCFM, shown in Figure 13, are significantly lower than those of other models across all time frames. In all cases, CCFM not only produces solutions with high accuracy but also yields samples that strictly satisfy the constraints. Figure 14 shows an example of the true solution field compared with generated ones from the models. Visually, the CCFM-generated sample consistently aligns with the ground truth, especially at early time steps where DPDE and FFM fail. PCFM achieves closer performance in the early steps yet exhibits larger magnitude discrepancies (lighter blue than the ground truth). Table 10 presents a comprehensive performance comparison of five different methods on the 2D Navier–Stokes equation solution generation task. Crucially, all results are reported as the mean performance with associated standard deviations, derived from 5 independent runs, demonstrating that the CCFM method achieves the best overall and most stable performance across the key error and conservation metrics.

### C.3 PHYSICS-INFORMED MOTION

Figure 15 illustrates generated trajectories under Moon gravity, where the proposed approaches align closely with the ground truth (red circles) compared to the baseline while ensuring high-fidelity generated samples.

# D    LLM USAGE

We used a large language model (LLM) for grammar correction and language polishing.

