# OpenReview forum: "Chance-constrained Flow Matching for High-Fidelity Constraint-aware Generation"
_ICLR.cc/2026/Conference — Submitted to ICLR 2026_

### Official Review · Reviewer_fztK · 2025-10-25

**Soundness:** 2
**Presentation:** 2
**Contribution:** 3
**Rating:** 4
**Confidence:** 3

**Summary:**

This paper presents Chance-constrained Flow Matching algorithm(CCFM)  aimed for a generation of samples given a certain constraint condition.  CCFM introduces the constraint indirectly by applying the chance-modulated constraint, which is then shown to be mathematically convertible to deterministic constraint in the form of euclidean projection.    By repeatedly applying this constraint projection at the time of the inference, the algorithm produces samples  that satisfy the constraint. The paper shows the efficacy of the method  on PDE and Molecular Docking.

**Strengths:**

###1
By "regularizing" the constraint via  probabilistic formulation, the method outperforms other generative models with constraint.

###2
The converstion of the probabilstic constraint to determistic projections are mathematically justified, along with the guarantee that the intermediate constraint would also make the finally generative samples feasible.

**Weaknesses:**

While the paper tackles an important problem and supports the algorithm is several theoretical guarantees,  the paper seems to lack several components necessary for clarity. Please see the Questions section.

Minor points: In (6), $min_x$ instead of $min_y$?   In (11), is the constraint defined interms of $x_t'$?

**Questions:**

###1 What is the mathematical goal of the model? What is CCFM guaranteed to generate? For a moment I thought the goal was to generate $P_{target}(\cdot | constraint)$, but I guess that the model generates $\pi_{constraint}$ #  $P_{target}$, the pushforward of the Constraint Projection operator to the target distribution?   Is there any theoretical guarantee for what CCFM is producing at the end?

###2 The probabistic parameter $\alpha$, after all, seems to be acting as the regularizer of the projection operator ; as the regularization strength, its scheduling seems vital in the training.  As a method, both a good heuristics and theoretical support seems warranted.

###3 The Projection operator seems to require "if" statement for each sample in most of the applications discussed here.  Wouldn't this pose a computational risk? It is being claimed that CCFM achieves faster runtime than PCFM, what is the essential computational gain?

###4 In section 4,  CCFM is motivated by the problem of previous works applying repeated projections to the noisy samples---however, the methods suffering from this problem do not seem to be compared against in the experimental section.

**Details Of Ethics Concerns:**

Nothing in Particular.

---

> ### Author Response · Authors · 2025-11-21
>
> Thank you for the helpful comments, in particular the acknowledgement of the **strong empirical performance** of our proposed CCFM and **its sound theoretical foundation.** We provide below our answers to your constructive questions.
>
> - **Q1: What is the mathematical goal of the model? What is CCFM guaranteed to generate? For a moment I thought the goal was to generate $P_{target}(·|constraint)$, but I guess that the model generates $\pi_{constraint}$ # $P_{target}$ , the pushforward of the Constraint Projection operator to the target distribution? Is there any theoretical guarantee for what CCFM is producing at the end?**
>
> We clarify that the mathematical goal of CCFM is to generate $\Pi$ # $P_{\theta}$, which is the pushforward of the learned distribution $P_{\theta}$ onto the feasible set $\{x : g(x) \le 0\}$ via the chance-constrained projection operator $\Pi$.  Since all training data satisfy the constraint, the target distribution $P_{target}$ is already supported on the feasible set, that is, $P_{\text{target}} = P_{\text{data}}(\cdot|\text{constraint}) = \pi_{constraint}$ # $P_{target}$. However, due to the nature of machine learning, the learned distribution $P_{\theta}$ always differs from the target distribution $P_{target}$.
>
> CCFM guarantees that all generated samples satisfy the constraint, as established in Corollary 2. Furthermore, Theorems 3 and 4 show that our framework preserves the geometry of the feasible set along the entire sampling path, ensuring that constraint enforcement does not distort the underlying structure.
>
> - **Q2: The probabistic parameter $\alpha$, after all, seems to be acting as the regularizer of the projection operator ; as the regularization strength, its scheduling seems vital in the training. As a method, both a good heuristics and theoretical support seems warranted.**
>
> We agree that probabilistic parameter $\alpha$ is vital in the projection operator. However, we would like to clarify that CCFM is a training-free method: this parameter is used only during inference, not during model training.
>
> The chance constraint is formulated as:
>
> $$\mathbb{P}_{\xi}(g(t^{-1}x_t - \xi) \le 0) \ge 1-\alpha$$,
>
> where the noise $\xi = \frac{1-t}{t}x_0$, and the confidence level is determined by the scheduler $\phi(t) = (t/2)^n$. As $t$ grows, the constraint becomes tighter. At $t=1$ the constraint becomes a deterministic projection, which ensures strict feasibility of the final samples (Corollary 2).
>
> A large $n$ keeps $\phi(t) $ near zero for most of the trajectory, so the sampler behaves like standard flow matching and applies projection mainly near the end; in the limit $n\to\infty$, this becomes pure post-processing. A small $n$ introduces projection early, and $n=0$ reduces to repeated projection.
>
> In practice, we choose a moderate $n$ based on task difficulty. Harder tasks may benefit from earlier corrections (smaller $n$), while easier tasks can rely more on the learned flow (larger $n$). The method is stable across a wide range of values: Tables 4 and 6 show that feasibility remains unchanged, and MMSE/SMSE vary only slightly. This confirms that the scheduler does not influence the final feasibility guarantee, which is enforced by the deterministic projection at $t=1$.

---

> ### Author Response · Authors · 2025-11-21
>
> - **Q3: The Projection operator seems to require "if" statement for each sample in most of the applications discussed here. Wouldn't this pose a computational risk? It is being claimed that CCFM achieves faster runtime than PCFM, what is the essential computational gain?**
>
> We would like to clarify that the "if" statement is implemented as a vectorized violation check (a Boolean mask) on the GPU. As a result, it introduces minimal computational costs and does not dominate the runtime.
>
> The computational gain of CCFM over PCFM comes from two sources: (1) CCFM eliminates the costly extrapolation–correction–interpolation loop required by PCFM. Instead, it applies the chance-constrained projection directly after each sampling step, avoiding additional integration overhead. (2) the parameter $\phi(t) $ the schedule of constraint enforcement. In many configurations, constraints only become active during the later stages of the sampling process. This avoids constraint enforcement during the early sampling phase, drastically reducing computation compared to PCFM’s always-on deterministic projection.
>
> - **Q4: In section 4, CCFM is motivated by the problem of previous works applying repeated projections to the noisy samples---however, the methods suffering from this problem do not seem to be compared against in the experimental section.**
>
> We sincerely thank the reviewer for this constructive suggestion.
>
> While we have compared our CCFM against the state-of-the-art baselines for each task, we agree that a comparison with standard Repeated-Projection (RP) methods is important. We provide the RP results, and the full comparison has been added to the revised draft. In addition, our CCFM consistently outperforms the repeated-projection method across all evaluated settings.
>
> - Results on Molecular Docking
>
>   | **#S** | **#Stp** | **Methods** | **PB Valid +L < 2Å (%)** | **PB Valid +A < 2Å (%)** |
>   | ------ | -------- | ----------- | ------------------------ | ------------------------ |
>   | **1**  | **2**    | CCFM        | **22.03**                | **35.88**                |
>   |        |          | RP          | 9.32                     | 14.69                    |
>   | **10** | **2**    | CCFM        | **31.36**                | **51.98**                |
>   |        |          | RP          | 21.19                    | 34.18                    |
>   | **20** | **2**    | CCFM        | **33.62**                | **51.41**                |
>   |        |          | RP          | 25.42                    | 37.85                    |
>   | **1**  | **10**   | CCFM        | **30.79**                | **58.19**                |
>   |        |          | RP          | 27.40                    | 42.66                    |
>   | **1**  | **20**   | CCFM        | **29.94**                | **57.06**                |
>   |        |          | RP          | 26.55                    | 40.68                    |
>   | **10** | **10**   | CCFM        | **35.88**                | **62.15**                |
>   |        |          | RP          | 34.75                    | 56.78                    |
>
> - Results on PDE Problems
>
> | **Problem**            | **Metric**                               | **CCFM** | **RP** |
> | ---------------------- | ---------------------------------------- | -------- | ------ |
> | **Reaction-Diffusion** | MMSE ($\times 10^{-2}$) $\downarrow$     | **3.3**  | 8.1    |
> |                        | SMSE ($\times 10^{-2}$) $\downarrow$     | **2.7**  | 8.5    |
> |                        | CV (IC) ($\times 10^{-2}$) $\downarrow$  | **0**    | **0**  |
> |                        | CV (CL) ($\times 10^{-15}$) $\downarrow$ | **9.5**  | 10.8   |
> | **Navier-Stokes**      | MMSE ($\times 10^{-1}$) $\downarrow$     | **0.8**  | 1.0    |
> |                        | SMSE ($\times 10^{-1}$) $\downarrow$     | **0.8**  | 0.9    |
> |                        | CV (IC) $\downarrow$                     | **0**    | **0**  |
> |                        | CV (CL) ($\times 10^{-15}$) $\downarrow$ | 3.9      | 4.1    |
>
>
>
> ---
>
> Thank you, again, for your time and feedback. We believe that the additional results and explanations included in our response have addressed the concerns presented, but we encourage you to follow-up on any points that remain unclear. Additionally, we encourage the reviewer to check the updated version of our submission, which has been extended to incorporate these additional results and discussions. We hope these revision will merit your revised support for this work!

---

> ### Author Response · Authors · 2025-11-26
> **Follow-up and clarification**
>
> As the discussion period is nearing its end, we wanted to ask **Reviewer fztK** if there are any follow-up points we can clarify.
>
> We believe we have responded to all questions and concerns raised, in addition to taking a few days to run the additional experiments necessary to demonstrate our points, all of which are incorporated in the revised version of the paper.
> \
> If there are no further points of clarification regarding the manuscript, we kindly ask that reviewer fztK consider increasing their score to reflect the improvements and clarifications we have provided. We are happy to continue to engage in discussion and answer any questions!

---

### Official Review · Reviewer_xbCA · 2025-10-31

**Soundness:** 3
**Presentation:** 3
**Contribution:** 2
**Rating:** 4
**Confidence:** 3

**Summary:**

The paper extends recent repeated projection schemes for training-free constrained generation via projection, to a chance-constrained stochastic optimization framework. This formal viewpoint leads to a practically-relevant algorithm with theoretical guarantees for relatively idealistic settings and positive experimental results on practically relevant problems with general complex constraint sets.

**Strengths:**

- The paper provides a new rigorous viewpoint on constrained generation by transporting standard understanding in stochastic optimization theory. I regard this connection and formalized problem as highly principled and valuable.

- The problem treated (i.e., constrained generation for flows/diffusion) is highly practically relevant and it is still arguably an open problem.

- The paper is well-structured, and well written.

- Both the theoretical analysis presented within sec. 4.3, and experiments within sec. 5 seem mostly convincing.

**Weaknesses:**

- (main concern) Sec. 4.2 explains that the presented opt. problem with probabilistic constraints is intractable and Prop. 1 reformulates it so that it becomes tractable via a deterministic reformulation. But it seems to me that Prop. 1 only tackles the case of linear/quadratic constraints, which arguably never capture typical constraints. I could not understand how the (arguably most common) case of non-convex constraints is managed. As of now, this seems to render the presented chance-constrained algorithmic machinery very limited.

- (main concern) The paper seems to propose an improved repeated-projection approach.  Nonetheless, reading the experiments sec. I could not clearly understand where to find a comparison with current repeated-projection schemes which do not leverage the chance-constrained formulation to determine the intermediate constraints. Where is it? Is it missing? I believe this experimental comparison would be essential to assess the gain of the core contribution of the paper, which is arguably the chance-constrained-based method as a way to improve repeated-projection schemes.

- I have fundamental doubts about the 'entire' repeated-projection approach. It seems to me that constraints are typically defined at the data level (e.g., g(x_0)). Due to the Markovian structure of the diffusion/flow process, a large amount of works [e.g., 1] interpret the sampling process as an MDP. This makes it possible to reduce constrained generation to constrained planning, a standard problem in RL/control, e.g. as done in [2] for the fine-tuning case. This standard RL/control viewpoint renders it possible to automatically derive intermediate deterministic (value) functions which maximization would lead to reward maximization at the last time-step. While the standard projection onto the constraint set C in previous repeated-projection works seems to me to make very little sense, this work arguably determines reasonable constraints for intermediate steps via the linear optimal transport viewpoint. But these constraints are probabilistic (and hence leading to intractable opt. problems), while methods exploiting the dynamic programming structure of the problem seem to lead to formulations corresponding to standard planning/RL tasks, which are arguably easier. Moreover, due to the control theoretic interpretation of classifier guidance [see 3], this logic extends also to classifier guidance given a proper weighting parameter of the 'penalty' term (which can be computed e.g. via Augmented Lagrangian schemes as in [2]). What is the authors opinion about this point?

- (writing, minor) I found the structure of presentation quite confusing. In particular, it seems to me that the chance-constrained viewpoint is leveraged as a way to develop algorithmic machinery to tackle a deterministic constraint problem. I would suggest to first introduce the formal problem on the data-level (i.e., last time-step constraint) and only afterwards introduce the chance-constrained framework implying probabilistic constraints at intermediate steps. Currently, I understood only late in the text that the data-level objective is deterministic rather than probabilistic. Since also a data-level probabilistic objective would make sense, I found this presentation quite misleading.

**References**

[1] Training Free Guided Flow Matching with Optimal Control, 2024.

[2] Constrained Molecular Generation via Sequential Flow Model Fine-Tuning, 2025.

[3] Variational Control for Guidance in Diffusion Models, 2025.

**Questions:**

I am happy to change my score if convincing answers to the posed questions are provided.

- Sec. 2 mentions that Classifier guidance (CG) schemes cannot provide feasibility guarantees. Could the authors clarify (formally) what they refer to with feasibility guarantee and why CG schemes cannot achieve it?

- How is optimization carried out for non-linear/quadratic constraints not leading to deterministic reformulations of intermediate probabilistic opt. problems?

- See questions within the Weaknesses sec. of the review.

---

> ### Author Response · Authors · 2025-11-21
>
> Thank you for your valuable feedback including acknowledging the **our new rigorous viewpoint on constrained generation**. We provide below our answers to your valuable questions:
>
> - **Q1: Could the authors clarify (formally) what they refer to with feasibility guarantee and why CG schemes cannot achieve it?**
>
> Thank you for that question.
>
> A **feasibility guarantee** is defined as the strict satisfiability of hard constraints. Formally, given a constraint function $g(x) \le 0$ (e.g., boundary conditions or physical laws), a generative model provides a feasibility guarantee if and only if the generated samples lie in the feasible set $\{x : g(x) \le 0\}$.
>
> We clarify that guidance-based methods, including Classifier Guidance (CG) and Classifier-Free Guidance (CFG), cannot provide such a guarantee because they act as **soft constraints**. Specifically, these methods correct the vector field ${v}_t(x)$ by a guidance term $B_t(x)$ [1]:
>
>  $$\tilde{v}_t(x) = v_t(x) + \omega \cdot B_t(x)$$,
>
> where $\tilde{v}_t$ is the perturbed velocity, $\omega$ is the guidance weight. For constraint satisfaction, $B_t(x)$ can be derived from the constraint violation (e.g., $B_t(x)  \propto  -\nabla g(x_t))$. From a distributional perspective, this modification can be interpreted as sampling from a reweighted density [2,3]:
>
> $$ \tilde{p}_\omega(x) \propto p(x) \exp\bigl(-\omega \cdot \max(0, g(x))\bigr), $$
>
> where $p(x)$ is the unconditional prior. As long as the prior $p(x)$ assigns non-zero probability mass to the infeasible region $\{x : g(x) > 0\}$ and the guidance weight $\omega$ is finite, the reweighted density $\tilde{p}_\omega(x)$ **remains strictly positive** in the infeasible region. Therefore, while such guidance can reduce the violation rate, it theoretically cannot ensure that $g(x) \le 0$ for all generated samples.
>
> - **Q2: How is optimization carried out for non-linear/quadratic constraints not leading to deterministic reformulations of intermediate probabilistic opt. problems?**
>
> Thank you for this insightful question.
>
> We agree that Proposition 1 leverages linear and quadratic forms to enable exact deterministic reformulations. It can cover a wide range of constraints in multiple domains. Notably, it successfully handles the nonlinear mass conservation laws in Reaction–Diffusion Equation and bond-length constraints in Molecular Docking (both non-convex), as utilized in prior works [4,5].
>
> For general non-linear/quadratic constraints, our CCFM framework remains applicable through two standard approaches:
>
> 1. **Local Linearization:** Similar to the Gauss-Newton projection used in prior work [5], general nonlinear constraints can be approximated via iterative local linearization.
> 2. **Stochastic Optimization:** A great number of established stochastic optimization techniques, such as Sample Average Approximation [6], Bernstein Approximation [7], or CVaR Approximation [8], can be employed. These methods can effectively address non-linear/quadratic constraints without altering our current framework.
>
> **Reviewer’s Additional Points**:
>
> > **A1**:  I could not understand how the (arguably most common) case of non-convex constraints is managed.
>
> Please refer to our response to Q2 for details.
>
> > **A2**: I believe this experimental comparison would be essential to assess the gain of the core contribution of the paper, which is arguably the chance-constrained-based method as a way to improve repeated-projection schemes.
>
> We sincerely thank the reviewer for this constructive suggestion.
>
> While we have compared our CCFM against the state-of-the-art baselines for each task, we agree that a comparison with standard Repeated-Projection (RP) methods is important. We provide the RP results, and the full comparison has been added to the revised draft. In addition, our CCFM consistently outperforms the repeated-projection method across all evaluated settings.

---

> ### Author Response · Authors · 2025-11-21
>
> - Results on Molecular Docking
>
>   | **#S** | **#Stp** | **Methods** | **PB Valid +L < 2Å (%)** | **PB Valid +A < 2Å (%)** |
>   | ------ | -------- | ----------- | ------------------------ | ------------------------ |
>   | **1**  | **2**    | CCFM        | **22.03**                | **35.88**                |
>   |        |          | RP          | 9.32                     | 14.69                    |
>   | **10** | **2**    | CCFM        | **31.36**                | **51.98**                |
>   |        |          | RP          | 21.19                    | 34.18                    |
>   | **20** | **2**    | CCFM        | **33.62**                | **51.41**                |
>   |        |          | RP          | 25.42                    | 37.85                    |
>   | **1**  | **10**   | CCFM        | **30.79**                | **58.19**                |
>   |        |          | RP          | 27.40                    | 42.66                    |
>   | **1**  | **20**   | CCFM        | **29.94**                | **57.06**                |
>   |        |          | RP          | 26.55                    | 40.68                    |
>   | **10** | **10**   | CCFM        | **35.88**                | **62.15**                |
>   |        |          | RP          | 34.75                    | 56.78                    |
>
> - Results on PDE Problems
>
> | **Problem**            | **Metric**                               | **CCFM** | **RP** |
> | ---------------------- | ---------------------------------------- | -------- | ------ |
> | **Reaction-Diffusion** | MMSE ($\times 10^{-2}$) $\downarrow$     | **3.3**  | 8.1    |
> |                        | SMSE ($\times 10^{-2}$) $\downarrow$     | **2.7**  | 8.5    |
> |                        | CV (IC) ($\times 10^{-2}$) $\downarrow$  | **0**    | **0**  |
> |                        | CV (CL) ($\times 10^{-15}$) $\downarrow$ | **9.5**  | 10.8   |
> | **Navier-Stokes**      | MMSE ($\times 10^{-1}$) $\downarrow$     | **0.8**  | 1.0    |
> |                        | SMSE ($\times 10^{-1}$) $\downarrow$     | **0.8**  | 0.9    |
> |                        | CV (IC) $\downarrow$                     | **0**    | **0**  |
> |                        | CV (CL) ($\times 10^{-15}$) $\downarrow$ | 3.9      | 4.1    |
>
>
>
> > **A3**: I have fundamental doubts about the 'entire' repeated-projection approach. It seems to me that constraints are typically defined at the data level (e.g., g(x_0)). Due to the Markovian structure of the diffusion/flow process, a large amount of works [e.g., 1] interpret the sampling process as an MDP. This makes it possible to reduce constrained generation to constrained planning, a standard problem in RL/control, e.g. as done in [2] for the fine-tuning case. This standard RL/control viewpoint renders it possible to automatically derive intermediate deterministic (value) functions which maximization would lead to reward maximization at the last time-step. While the standard projection onto the constraint set C in previous repeated-projection works seems to me to make very little sense, this work arguably determines reasonable constraints for intermediate steps via the linear optimal transport viewpoint. But these constraints are probabilistic (and hence leading to intractable opt. problems), while methods exploiting the dynamic programming structure of the problem seem to lead to formulations corresponding to standard planning/RL tasks, which are arguably easier. Moreover, due to the control theoretic interpretation of classifier guidance [see 3], this logic extends also to classifier guidance given a proper weighting parameter of the 'penalty' term (which can be computed e.g. via Augmented Lagrangian schemes as in [2]). What is the authors opinion about this point?
>
> We thank the reviewer for this insightful point.
>
> We agree that the RL/control perspective is valuable for guiding generative models. However, our work addresses a distinct problem setting: enforcing **hard** constraints on a **pretrained** flow model with strict feasibility guarantees, rather than optimizing a reward signal.
>
> Works such as [7] and [9] mentioned by the reviewer bias a pretrained model in a training-free way, but their objective is guided generation. They adjust the sampling path to lower the trajectory cost, not to ensure that all final samples satisfy a feasible set. As a result, there is no formal guarantee that $g(x) \le 0$ holds. Also, these methods require additional sampling steps to backpropagate control signals. Our method introduces a chance constrained projection operator at each sampling step directly to provide the feasibility guarantee.
>
> Reference [8] mentioned by the reviewer fine-tunes the model with an augmented Lagrangian. This requires extra data and a training phase. Also, it is not suitable for unseen constraint configurations. Furthermore, it cannot guarantee because the learned model is not perfect.

---

> ### Author Response · Authors · 2025-11-21
>
> In summary, RL and control based methods [1–3] provide useful guided generation tools, but they either need added sampling steps or do not provide guarantees on feasibility. Our method contributes a training-free scheme that enforces constraints on pretrained generative models with a feasibility guarantee.
>
>
>
> > **A4**: (writing, minor) I found the structure of presentation quite confusing. In particular, it seems to me that the chance-constrained viewpoint is leveraged as a way to develop algorithmic machinery to tackle a deterministic constraint problem. I would suggest to first introduce the formal problem on the data-level (i.e., last time-step constraint) and only afterwards introduce the chance-constrained framework implying probabilistic constraints at intermediate steps. Currently, I understood only late in the text that the data-level objective is deterministic rather than probabilistic. Since also a data-level probabilistic objective would make sense, I found this presentation quite misleading.
>
> We agree that the presentation can be clearer. In the revised draft, we have stated at the beginning that the data-level constraint is deterministic, and then explain that the chance-constrained formulation
>
> ---
>
> Thank you, again, for your time and feedback. We believe that the additional results and explanations included in our response have addressed the concerns presented, but we encourage you to follow-up on any points that remain unclear. Additionally, we encourage the reviewer to check the updated version of our submission, which has been extended to incorporate these additional results and discussions. We hope these revision will merit your revised support for this work!
>
> **References**
>
> [1] Ajay, Anurag, et al. "Is Conditional Generative Modeling all you need for Decision Making?." *The Eleventh International Conference on Learning Representations*.
>
> [2] Carvalho, Joao, et al. "Motion planning diffusion: Learning and planning of robot motions with diffusion models." *2023 IEEE/RSJ International Conference on Intelligent Robots and Systems (IROS)*. IEEE, 2023.
>
> [3] Shaoul, Yorai, et al. "Multi-Robot Motion Planning with Diffusion Models." *The Thirteenth International Conference on Learning Representations*.
>
> [4] Cheng, Chaoran, et al. "Gradient-Free Generation for Hard-Constrained Systems." *The Thirteenth International Conference on Learning Representations*.
>
> [5] Corso, Gabriele, et al. "Composing unbalanced flows for flexible docking and relaxation." *The Thirteenth International Conference on Learning Representations*. 2025.
>
> [6] Utkarsh, Utkarsh, et al. "Physics-Constrained Flow Matching: Sampling Generative Models with Hard Constraints." The Thirty-Ninth Annual Conference on Neural Information Processing Systems.
>
> [7] Luedtke, James, and Shabbir Ahmed. "A sample approximation approach for optimization with probabilistic constraints." *SIAM Journal on Optimization* 19.2 (2008): 674-699.
>
> [8] Nemirovski, Arkadi, and Alexander Shapiro. "Convex approximations of chance constrained programs." *SIAM Journal on Optimization* 17.4 (2007): 969-996.
>
> [9] Chen, Wenqing, et al. "From CVaR to uncertainty set: Implications in joint chance-constrained optimization." *Operations research* 58.2 (2010): 470-485.

---

> ### Author Response · Authors · 2025-11-26
> **Follow-up and clarification**
>
> As the discussion period is nearing its end, we wanted to ask **Reviewer xbCA** if there are any follow-up points we can clarify.
>
> We believe we have responded to all questions and concerns raised, in addition to taking a few days to run the additional experiments necessary to demonstrate our points, all of which are incorporated in the revised version of the paper.
> \
> If there are no further points of clarification regarding the manuscript, we kindly ask that reviewer xbCA consider increasing their score to reflect the improvements and clarifications we have provided. We are happy to continue to engage in discussion and answer any questions!

---

### Official Review · Reviewer_1myh · 2025-11-01

**Soundness:** 2
**Presentation:** 3
**Contribution:** 2
**Rating:** 2
**Confidence:** 2

**Summary:**

This paper introduces Chance-Constrained Flow Matching (CCFM), a training-free modification of flow matching designed to enforce hard constraints during sampling. The method reformulates the standard projection step as a chance-constrained program, allowing constraints to be satisfied with high probability on noisy intermediate states while preserving fidelity to the clean target distribution. Theoretical results claim equivalence between intermediate and final projections under an optimal transport path assumption, and tractable Gaussian surrogates are derived for linear and quadratic constraints. Experiments are presented on two scientific domains: molecular docking and PDE solution generation, showing improved feasibility and efficiency over existing baselines.

**Strengths:**

### **Strengths**

- The idea of combining flow matching with chance-constrained programming is conceptually interesting and introduces a probabilistic treatment of feasibility within generative flows, which is novel and potentially useful for scientific modeling.
- The theoretical exposition is clear and internally consistent under its assumptions, and the connection between intermediate and final feasible sets is elegant (Theorem 4).
- The method is simple to implement, training-free, and shows promising qualitative results on molecular docking and PDE benchmarks, indicating some practical value.

**Weaknesses:**

### **Weaknesses**

- The experimental evaluation is limited in scope and depth. Only two scientific domains are tested, each with narrow baselines and minimal ablation. The probability scheduler, risk level, and theoretical assumptions (convexity, linearity) are not systematically validated. Since ICLR emphasizes experimental rigor and breadth, the current empirical evidence is insufficient to support the paper’s general claims.
- The claimed equivalence between noisy and clean projections relies on restrictive assumptions that are unlikely to hold in realistic nonconvex cases like docking, yet no empirical study quantifies the resulting approximation error or its impact on feasibility.
- The docking and PDE comparisons lack strong baselines and statistical rigor. Metrics are presented without confidence intervals or robustness analyses, and the reported efficiency gains are small or inconsistent across tasks, making it difficult to assess the true advantage of CCFM.

**Questions:**

### **Questions**

Besides the implied ones based on the weaknesses above:

- Do the authors plan to test on other classes of constraints to demonstrate broader generality?
- Can the authors provide sensitivity analyses for the probability scheduler and risk level across both domains?

---

> ### Author Response · Authors · 2025-11-21
>
> Thank you for the helpful review, in particular the acknowledgement of the **conceptual novelty** of our approach in combining flow matching with chance-constrained programming. We provide below our answers to your insightful questions.
>
> - **Q1: Do the authors plan to test on other classes of constraints to demonstrate broader generality?**
>
> Thanks for the helpful suggestion!
>
> Our current experiments already include a wide range of **convex constraints** (such as the initial conditions for the Reaction–Diffusion, and the global mass conservation law in Navier–Stokes equations) and **nonconvex constraints** (such as the nonlinear mass conservation law in Reaction–Diffusion, and the bond-length, bond-angle, steric-clash, and protein–ligand contact constraints in molecular docking). Across these settings, our method *consistently improves both feasibility and fidelity when compared with existing state-of-the-art methods*.
>
> We also agree that adding more types of constraints would further support the generality of our method. Specifically, we consider the physics-informed motion task considered in [1]. We generate six-frame (4 seconds) video sequences in which a 2D object falls under Earth's gravity from a random initial in-frame height. An unconditional flow-matching model is trained to generate these trajectories. At the sampling phase, we consider a challenging out-of-distribution generation task, which instead imposes the Moon’s gravity as the physical constraint governing the object’s motion.
>
> To evaluate different constraint-handling strategies (unconstrained generation, repeated projection [1], and CCFM), we generate 100 samples using each method and compute the mean-squared error between the generated object positions and the analytically expected positions under Moon gravity (given the initial height). The results are summarized in the following table.
>
> |                                | Unconstrained | Repeated Projection [1] | CCFM   |
> | ------------------------------ | ------------- | ----------------------- | ------ |
> | Constraint Violation (MSE) (↓) | 278.46        | 0.8333                  | 0.0100 |
> | FID               (↓)          | 16.985        | 27.184                  | 26.767 |
>
> These metrics show that CCFM achieves the lowest constraint violation and a better FID score than repeated projection and unconstrained models. For the constrained generation task, where physical consistency is essential, CCFM delivers the most accurate and reliable results.
>
> - **Q2: Can the authors provide sensitivity analyses for the probability scheduler and risk level across both domains?**
>
> Thanks for that question!
>
> Appendix C already reports a sensitivity analysis of the probability scheduler on the PDE tasks. We observe that changing the probability scheduler has almost no effect on constraint satisfaction. The changes in MMSE and SMSE are also small and stay within a narrow range, showing that the method is robust to the choice of scheduler.
>
> Similarly, we conducted a sensitivity analysis for the probability scheduler on the molecular docking task. As observed with the PDE tasks, the performance remains stable across different values of $n$, indicating that the method is also robust to scheduler variations in the molecular docking domain.
>
> | Metric                | CCFM(0.7) | CCFM(0.9) | CCFM(1.0) | CCFM(1.1) | CCFM(1.3) |
> | :-------------------- | :-------: | :-------: | :-------: | :-------: | :-------: |
> | PB Valid (**+L <2Å**) |   29.85   |   30.16   |   30.79   |   30.48   |   30.48   |
> | PB Valid (**+A <2Å**) |   58.19   |   57.85   |   58.19   |   58.19   |   58.88   |
>
> In addition, to provide statistically meaningful results, we report the mean and standard deviation over 5 independent runs for both domains. These statistics show that CCFM consistently matches or outperforms the baselines, and that the performance gains are stable rather than due to random variation.
>
> For the molecular docking task, CCFM achieves the best or tied-best performance on most key metrics:
>
> | Methods  | L-RMSD < 1 Å (%) | L-RMSD < 2 Å (%) | A-RMSD < 2 Å (%) | PB Valid All (%) | PB Valid +L < 2 Å (%) | PB Valid +A < 2 Å (%) |
> | -------- | ---------------- | ---------------- | ---------------- | ---------------- | --------------------- | --------------------- |
> | CCFM     | **7.80 ± 0.31**  | **32.57 ± 0.57** | 72.27 ± 0.69     | **79.15 ± 0.41** | **30.45 ± 0.68**      | **58.53 ± 0.49**      |
> | FlexDock | 7.45 ± 0.43      | 31.92 ± 0.81     | **72.65 ± 0.81** | 58.07 ± 1.24     | 24.57 ± 1.13          | 40.96 ± 1.38          |
> | RP       | **7.80 ± 0.34**  | 32.19 ± 0.79     | 72.26 ± 0.99     | 56.48 ± 0.72     | 27.68 ± 0.81          | 42.93 ± 0.85          |

---

> ### Author Response · Authors · 2025-11-21
>
> For the 2D Navier–Stokes PDE task, CCFM also gives the lowest MMSE/SMSE while maintaining strict or near-strict constraint satisfaction:
>
> | Problem       | Metric              | CCFM (ours)     | PCFM           | DiffusionPDE      | FFM              | RP          |
> | ------------- | ------------------- | --------------- | -------------- | ----------------- | ---------------- | ----------- |
> | Navier–Stokes | MMSE × 10⁻¹ (↓)     | **0.79 ± 0.03** | 1.67 ± 0.09    | 1.44 ± 0.10       | 1.56 ± 0.16      | 1.08 ± 0.08 |
> |               | SMSE × 10⁻¹ (↓)     | **0.81 ± 0.11** | 1.53 ± 0.18    | 9.21 ± 0.39       | 9.57 ± 0.53      | 0.97 ± 0.25 |
> |               | CV (IC) (↓)         | **0**           | **0**          | 0.12 ± 0.03       | 0.13 ± 0.08      | **0**       |
> |               | CV (CL) × 10⁻¹⁵ (↓) | 3.9 ± 0.01      | **3.4 ± 0.02** | 1.2 ± 0.13 × 10¹⁰ | 1.2 ± 0.12 × 10⁹ | 4.1 ± 0.04  |
>
> The corresponding results have also been added to Appendix C of the revised paper.
>
> **Reviewer’s Additional Points**:
>
> > **A1**: The experimental evaluation is limited in scope and depth.
>
> We have compared against the current state-of-the-art methods in two different domains. We also agree that expanding the scope and depth of the evaluation would further strengthen the work. Please refer to our response to Q1 for details.
>
> > **A2**: The claimed equivalence between noisy and clean projections relies on restrictive assumptions that are unlikely to hold in realistic nonconvex cases.
>
> The geometric relation between intermediate and final projections is proved under a convexity assumption, which is standard and widely used in prior work. Empirically, our results on molecular docking show that this relation remains accurate even in highly nonconvex settings.
>
> The feasibility theorem does not rely on convexity. Feasibility therefore does not depend on the type of constraint, which is also supported by our experiments.
>
> > **A3**: The docking and PDE comparisons lack strong baselines and statistical rigor
>
> We already compare against the state-of-the-art methods in both domains and observe consistent improvements across tasks and metrics. We agree that additional experiments and statistical evidence will strengthen the evaluation. Please refer to our response to Q1 for details.
>
> ---
>
> Thank you, again, for your time and feedback. We believe that the additional results and explanations included in our response have addressed the concerns presented, but we encourage you to follow-up on any points that remain unclear. Additionally, we encourage the reviewer to check the updated version of our submission, which has been extended to incorporate these additional results and discussions. We hope these revision will merit your revised support for this work!
>
> **References**
>
> [1] Christopher, Jacob K., Stephen Baek, and Nando Fioretto. "Constrained synthesis with projected diffusion models." *Advances in Neural Information Processing Systems* 37 (2024): 89307-89333.

---

> ### Author Response · Authors · 2025-11-26
> **Follow-up and clarifications**
>
> As the discussion period is nearing its end, we wanted to ask **Reviewer 1myh** if there are any follow-up points we can clarify.
>
> We believe we have responded to all questions and concerns raised, in addition to taking a few days to run the additional experiments necessary to demonstrate our points, all of which are incorporated in the revised version of the paper.
> \
> If there are no further points of clarification regarding the manuscript, we kindly ask that reviewer 1myh consider increasing their score to reflect the improvements and clarifications we have provided. We are happy to continue to engage in discussion and answer any questions!

---

### Meta-Review · Area_Chair_hSiE · 2026-01-10

**Summary:**

First, regarding the *authors* concerns:

- 1myh has some elements that may raise eyebrows about this ("Since ICLR, ..."), but note that the tool was confidence 2, and the review wasn't *completely* devoid of concerns. While some of those comments are generic, it's not like e.g. asking for error bars is an obviously AI generated or contentless remark, so I'm not really inclined to raise that further.
- Regarding Reviewer ftzk: It feels to me like correcting this misconception in the rebuttal is the correct approach here, which you have done. Many reviewers have many misconceptions, this doesn't seem to me to rise to the level of actionable.

The reviewers raised a few methodological concerns, including positioning and comparison to standard RP, sensitivity analysis and error bars, and a number of just points of confusion about the metholodogy, structure of the paper, precise definition of constraints being enforced, etc.

**Reviewer Concerns:**

The authors:
- Added repeated projection results as requested.
- Added a sensitivity analysis for probability scheduling
- Made efforts to address methodological questions and points e.g. raised by xbCA
- Clarified some misconceptions for ftzk and added a comparison to RP

**Reviewer Scores:**

Hard to say. I do think the authors in many cases do a decent job responding to the reviewers. However, what is also clear to me is that the general level of reviewer confusion was significant, and there were a lot of questions beyond technical concerns about organization, clarification of the constraints the method is designed to tackle, and general structure of the paper. My guess is that a version of the manuscript that incorporates the author feedback would likely be received more positively by another review panel; however it's hard to guess that *all* reviewers would increase their scores by enough to make the difference here.

---

### Decision · Program_Chairs · 2026-01-26

Reject